# EXCELFORMER: MAKING NEURAL NETWORK EXCEL IN SMALL TABULAR DATA PREDICTION

## ABSTRACT

Data organized in tables are omnipresent in real-world applications. Despite their strong performance on large-scale datasets, deep neural networks (DNNs) perform inferior on small-scale tabular data, which hinders the wider adoption of DNNs across domains. In this paper, we propose a holistic framework comprising a novel neural network architecture called EXCELFORMER and two data augmentation approaches, which achieves high-precision prediction for supervised classification and regression tasks, particularly on small-scale tabular datasets. The core component of EXCELFORMER is a novel *semi-permeable attention* coupled with a special initialization, which explicitly diminishes the impacts of uninformative features, thereby improving data-efficiency. The methodology insight behind two tabular data augmentation approaches, FEAT-MIX and HID-MIX, is to increase the training samples in a way accommodating the inherent irregularities of data patterns. Comprehensive experiments on diverse small-scale tabular datasets show that, our EXCELFORMER consistently and substantially outperforms previous works, with no noticeable dataset type preference. Remarkably, we find the superiority of EXCELFORMER extends to large datasets as well.

## 1 INTRODUCTION

Tabular data is ubiquitous and plays a critical role in real-world applications, spanning diverse domains such as medical trial prediction (Wu et al., 2022), market prediction (Wang et al., 2017), and financial risk forecasting (Kim et al., 2020a). While deep neural networks (DNNs) have been firmly established as state-of-the-art approaches in various domains, their performance on tabular datasets, particularly on smaller-scale ones, still exhibits limitations in performance (Grinsztajn et al., 2022). Besides, it is widely observed that no previous approaches consistently achieve strong performances across various tabular data prediction tasks, showcasing the opportunity to construct an effective neural networks for tabular data prediction. These issues present significant challenges and bottlenecks in the broader adoption of neural networks on tasks that involve tables.

Why do DNNs underperform on tabular datasets, especially the small-scale ones? An obvious reason is that DNNs have a larger hypothesis space, and thus require a large amount of data to achieve strong performance. Another answer is: DNNs' learning procedure exhibits rotational invariance property, which results in its worst-case sample complexity grows at least linearly in the number of informative features (Ng, 2004). Furthermore, tabular data inherently lack rotational invariance, and thus the rotational invariance characteristic of DNNs does not provide advantages in this context. Conversely, they necessitate a larger number of samples in learning to discern the original orientation of features. On the contrary, some conventional approaches (*e.g.*, GBDTs) are not rotationally invariant (Ng, 2004), demonstrating greater efficiency on tabular data. Hence, in cases of limited availability of data, mitigating informative feature interference or breaking DNNs' rotational invariance nature presents an opportunity to enhance their performance.

To overcome such limitation of DNNs, our approach draws on insights from two key perspectives:

(P1) Improving DNNs' data-efficiency. Due to the rotational invariance property, mitigate DNNs' emphasis on less informative features or break their rotational invariance reduces the data requirement.

(P2) Increasing training samples. Augmenting the training dataset presents the most direct approach to addressing the challenge of inadequate training samples.

Motivated by this, we introduce a novel model called EXCELFORMER, which is constructed based on the Transformer architecture (Vaswani et al., 2017). EXCELFORMER incorporates a key component called *semi-permeable attention* module (SPA), which selectively permits more informative features to gather information from less informative ones. This results in a noticeably reduced influence of less informative features, leading to decreased data requirement according to (P1). Specially, we introduce a new *interaction-attenuation initialization* approach to enhance the learning of SPA. This initialization approach sets SPA's parameters with minimal values, effectively attenuating tabular data feature interactions during the initial stages of training. Consequently, this approach initializes EXCELFORMER as a more non-rotationally invariant model (disregarding the fully connection layer for target prediction), which helps to break the rotational invariance.

To expand the training datasets in accordance with (P2), we introduce two interpolation-based data augmentation approaches for tabular data: FEAT-MIX and HID-MIX. Interpolation-based data augmentation approaches, such as Mixup (Zhang et al., 2018) and its variants (Verma et al., 2019; Uddin et al., 2020), have demonstrated their effectiveness in computer vision tasks. However, because the target functions of tabular data are often irregular (Grinsztajn et al., 2022), simple interpolation methods, like Mixup, tend to regularize DNNs to behave linearly in-between training examples (Zhang et al., 2018), which is often in conflict with the irregular feature-target relation. Therefore, they often fall short of improving and even degrading the performance of DNNs (as evidenced by the empirical results as shown in Fig. 4). In contrast, our FEAT-MIX and HID-MIX approaches avoid such conflicts and respectively encourage DNNs to learn independent feature transformations and to conduct sparse feature interactions.

**Technical Contributions:** (i) We introduce a novel model called EXCELFORMER, incorporating a unique *semi-permeable attention* module alongside a specialized *interaction-attenuation initialization*, which effectively resolving the conflicts arising from the rotational invariance inherent to DNNs and the rotational variance characteristic of tabular data. (ii) Additionally, our newly proposed data augmentation approaches for tabular data, HID-MIX and FEAT-MIX, generate training samples to supplement samples in a way accommodating the irregular feature-target relation of tabular data.

**Practical Contributions:** Experiments across various real-world small datasets confirm the superiority of our proposed EXCELFORMER over GBDTs and previous deep learning approaches. Notably, (i) our EXCELFORMER, even with pre-defined hyperparameters, outperforms state-of-the-art methods with fine-tuned hyperparameters, reducing training time consumption. (ii) The performance of EXCELFORMER remains consistently best across subgroups of tabular datasets (e.g., classification datasets), showing no overt dataset type preference. (iii) Surprisingly, the superiority of EXCELFORMER extends to larger datasets, making it a versatile choice for tabular data prediction.

## 2 EXCELFORMER

In this section, we present the workflow of EXCELFORMER, as illustrated in Fig. 1. EXCELFORMER includes the following key components: 1) The embedding layer featurizes and embeds tabular features to token-level embeddings; 2) token-level embeddings were alternately processed by the newly proposed *semi-permeable attention* module (SPA) and gated linear units (GLUs). 3) Finally, a prediction head constructed by two fully connection layers was used to predict the final target. In the following, we will introduce the novel *semi-permeable attention* with the interaction attenuated initialization first and then the rest part of EXCELFORMER.

### 2.1 SEMI-PERMEABLE ATTENTION

As stated in (Ng, 2004), less informative features make minor contributions on target prediction but still necessitate at least a linear increase in the requirement for training samples to learn how to "ignore" them. Besides, DNNs may not generalize well if trained on insufficient amounts of data, and less informative features may introduce excessive noise and impede prediction. Given the limited availability of data on small-scale tabular datasets, our idea is to incorporate an inductive bias into the self-attention mechanism, which selectively restricts the impacts of a feature to only those that are less informative, thereby reducing the overall impact of uninformative features on prediction outcomes. We propose a *semi-permeable attention* module (SPA), which is defined by:

$$z' = \text{softmax}(\frac{(zW_q)(zW_k)^T \oplus M}{\sqrt{d}})(zW_v), \tag{1}$$

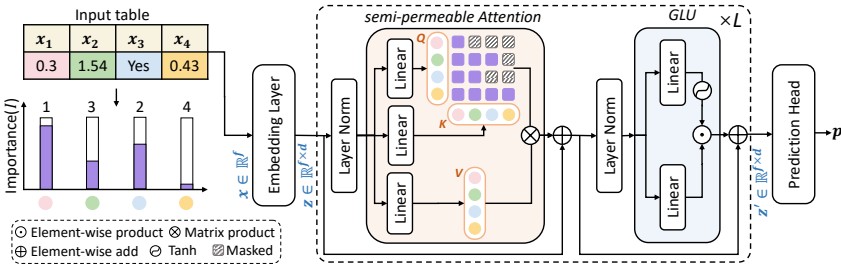

Figure 1: An illustration of our proposed EXCELFORMER model.

where $z \in \mathbb{R}^{f \times d}$ is the input embeddings and $z'$ the output embeddings, $W_q, W_k, W_v \in \mathbb{R}^{d \times d}$ are all learnable matrices, and $\oplus$ is element-wise addition. $M \in \mathbb{R}^{f \times f}$ is an unoptimizable mask, where the element at the $i$-th row and $j$-th column is defined by:

$$M[i,j] = \begin{cases} -\infty & I(\mathbf{f}_i) > I(\mathbf{f}_j) \\ 0 & I(\mathbf{f}_i) \le I(\mathbf{f}_j) \end{cases} \qquad (2)$$

The function $I(\cdot)$ represents a measure of feature importance, and we use the 'mutual information' metric in this paper (see Appendix F for details). If the feature $\mathbf{f}_i$ is more informative compared to $\mathbf{f}_j$, $M[i,j]$ is set $-\infty$ (we use $-10^5$ in implementation) and thus the $(i,j)$ grid on the attention map is masked. It prevents the transfer of the token embedding for the feature $\mathbf{f}_j$ to the token representing $\mathbf{f}_i$. In this way, only more informative features are permitted to propagate information to the less informative ones, and the reverse is not allowed. By doing so, SPA still maintains interaction pathways between any two features while constraining the impacts of less informative ones. Intuitively, when training samples are insufficient, some feature interactions conducted by the model may be sub-optimal, as vanilla self-attention was proved data-inefficient (Touvron et al., 2021a). When using SPA, it can avoid the excessive impacts of a noisy feature on prediction outcomes in case some associated interaction pathways are ill-suited. In practice, SPA is extended to a multi-head self-attention version, with 32 heads by default.

**Interaction Attenuated Initialization.** Since non-rotationally invariant algorithms have been recognized as more efficient learners for tabular data, we present a tailored initialization approach for our *semi-permeable attention* to ensure that EXCELFORMER starts as a largely non-rotationally invariant model. Notably, removing all self-attention operations from a Transformer model, features are processed individually, which makes the Transformer model nearly non-rotationally invariant (if we set aside the full connection layers that fuse features for target prediction). Concurrently, prior researches have evidenced the indispensable role of feature interactions (e.g., through self-attention) in Transformer-based models on tabular data (Gorishniy et al., 2021; Yan et al., 2023). By integrating these insights, our proposed *interaction attenuated initialization* scheme initially dampens the impact of SPA during the early stages of training, allowing essential feature interactions progressively grow under the driving force of the data.

Our *interaction attenuated initialization* scheme is built upon the commonly used He's initialization (He et al., 2015) or *Xavier* initialization (Glorot & Bengio, 2010), by rescaling the variance of an initialized weight $w$ with $\gamma$ ($\gamma \to 0^+$) while keeping the expectation at 0:

$$\mathrm{Var}(w) = \gamma \mathrm{Var}_{\mathrm{prev}}(w), \qquad (3)$$

where $\mathrm{Var}_{\mathrm{prev}}(w)$ denotes the weight variance used in the He's initialization and *Xavier* initialization. In this work, we set $\gamma = 10^{-4}$. To reduce the impacts of SPA, we apply Eq. (3) to all the parameters in the SPA module. Thus, EXCELFORMER works like a non-rotationally invariant model initially.

Actually, for a module with an additive identity shortcut like $y = \mathcal{F}(x) + x$, our initialization approach attenuates the sub-network $\mathcal{F}(x)$ and satisfies the property of *dynamical isometry* (Saxe et al., 2014) for better trainability. Some previous work (Bachlechner et al., 2021; Touvron et al., 2021b) suggested to rescale the $\mathcal{F}(x)$ path as $y = \eta \mathcal{F}(x) + x$, where $\eta$ is a learnable scalar initialized as 0 or a learnable diagonal matrix whose elements are of very small values. Different from these methods, our attenuated initialization approach directly assigns minuscule values to the weights during initialization. Our approach is better suited for the flexible learning of whether each feature interaction pathway should be activated or not, thereby achieving sparse attention.

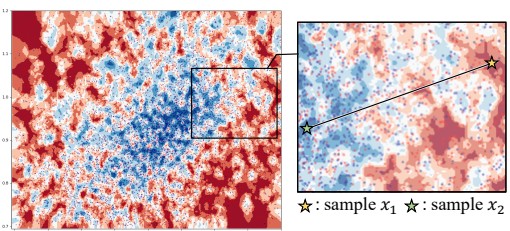

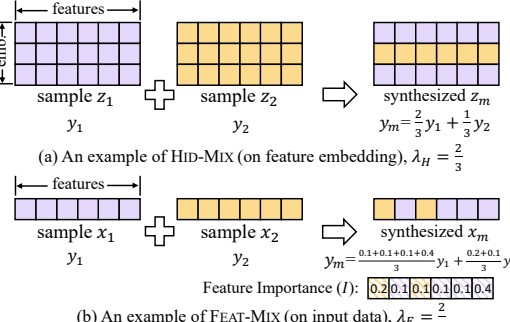

Figure 2: $k$NN ($k = 8$) decision boundaries with 2 key features of a zoomed-in part of the Higgs dataset. Convex combinations by vanilla Mixup (points on the black line) of 2 samples $x_1$ and $x_2$ may conflict with irregular category boundaries.

Figure 3: Examples for the HID-MIX and FEAT-MIX, where "emb." means "embeding" dimension.

## 2.2 OTHER COMPONENTS OF EXCELFORMER

**Feature Pre-processing.** Features are pre-processed before feeding into EXCELFORMER. The numerical features are normalized and the categorical features are converted into numerical ones using the CatBoost Encoder implemented with the *Sklearn* Python package [1]. This step performs similar to previous works (*e.g.*, Feature-Token Transformer (Gorishniy et al., 2021)).

**Embedding Layer.** The embedding layer transforms the input features into initial feature embedding $z^{(0)} \in \mathbb{R}^{f \times d}$ ($f$ and $d$ are the amount of feature and embedding dimension, respectively). The embedding $\mathbf{z}_i \in \mathbb{R}^d$ of a processed scalar feature $\mathbf{f}_i$ is computed, by:

$$\mathbf{z}_i = \tanh\left(\mathbf{f}_i W_{i,1} + b_{i,1}\right) \odot \left(\mathbf{f}_i W_{i,2} + b_{i,2}\right), \tag{4}$$

where $W_{i,1}, W_{i,2} \in \mathbb{R}^{1 \times d}$ and $b_{i,1}, b_{i,2} \in \mathbb{R}^d$ are learnable parameters. Then, the initial feature embedding $z^{(0)}$ are obtained by stacking $\mathbf{z}_i$, as $z^{(0)} = [\mathbf{z}_1, \mathbf{z}_2, \mathbf{z}_3, \ldots, \mathbf{z}_f]^T$.

**Gated Linear Unit Module.** In addition to the SPA module, another key component of EXCELFORMER is the Gated Linear Unit (GLU). As discussed in (Shazeer, 2020), replacing the point-wise feed-forward net (FFN) with the GLU module can enhance the Transformer. In our case, using GLU, an attentive module, can facilitate the learning of the irregular feature-target functions (Grinsztajn et al., 2022) on tabular datasets (also as shown in Fig. 2). Diverging from the standard GLU architecture, we employ the "tanh" activation in lieu of the "sigmoid" activation for better optimization properties (LeCun et al., 2002), as:

$$z' = \tanh\left(\texttt{Linear}_1(z)\right) \odot \texttt{Linear}_2(z), \tag{5}$$

where $\texttt{Linear}_1$ and $\texttt{Linear}_2$ are applied onto the embedding dimension of $z$, $\odot$ denotes element-wise product. Please note that both the vanilla FFN and GLU employ two fully connection layers, resulting in similar computational costs. The SPA and GLU modules are alternately stacked to form the core structure of the EXCELFORMER model, as shown in Fig. 1.

**Prediction Head.** The prediction head is directly applied to the output of the topmost GLU module, which contains two fully connection layers to separately compress the information along the token embeddings and fuse the information from features, by:

$$p = \phi(\texttt{Linear}_d(\text{P-ReLU}(\texttt{Linear}_f(z^{(L)})))), \tag{6}$$

where $z^{(L)}$ is the input, $W_f \in \mathbb{R}^{f \times C}$ and $b_f \in \mathbb{R}^C$. For multi-classification task, $C$ is the amount of target categories and $\phi$ indicates "softmax". For regression and binary classification tasks, then $C = 1$ and $\phi$ is *sigmoid*. The fully connection layer $\texttt{Linear}_f$ and $\texttt{Linear}_d$ are applied along and the feature dimension and the embedding dimension of $z^{(L)}$, respectively.

## 3 DATA AUGMENTATION

As mentioned in (P2), a straightforward approach to tackle data insufficiency is to create training samples. While Mixup (Zhang et al., 2018) regularizes DNNs to favor linear behaviors between

---

[1]https://contrib.scikit-learn.org/category_encoders/catboost.html

samples and stands as one of the most effective data augmentation methods in computer vision, empirical evidence suggests that it does not perform optimally on tabular datasets (*e.g.*, see Fig. 4). This discrepancy may be due to the conflict between the model's linear behavior and the irregularity of target functions, as intuitively illustrated in Fig. 2. To address this challenge, we introduce two Mixup variants, HID-MIX and FEAT-MIX, which mitigate the conflicts in creating samples.

**HID-MIX.** Our HID-MIX is applied to the token-level embeddings after the input samples have been processed by the embedding layer, along with their corresponding labels. It randomly exchanges some embedding "dimensions" between two samples (please refer to Fig. 3(a)). Let $z_1^{(0)}, z_2^{(0)} \in \mathbb{R}^{f \times d}$ be the token-level embeddings of two randomly selected samples, with $y_1$ and $y_2$ denoting their respective labels. A new sample represented as a token-label pair $(z_{\mathrm{m}}^{(0)}, y_{\mathrm{m}})$ is synthesized by:

$$\begin{cases} z_{\mathrm{m}}^{(0)} = S_H \odot z_1^{(0)} + (\mathbb{1}_H - S_H) \odot z_2^{(0)}, \\ y_{\mathrm{m}} = \lambda_H y_1 + (1 - \lambda_H) y_2, \end{cases} \tag{7}$$

where the matrix $S_H$ is of size $f \times d$ and is formed by stacking $f$ identical $d$-dimensional binary vectors denoted as $s_h$: $S_H = [s_h, s_h, \ldots, s_h]^T$. $s_h$ consists of $\lfloor \lambda_H \times d \rfloor$ randomly selected elements set to 1 and the rest elements set to 0. The scalar coefficient $\lambda_H$ for labels is sampled from the $\mathcal{B}eta(\alpha_H, \alpha_H)$ distribution, where $\alpha_H$ is a hyper-parameter. $\mathbb{1}_H$ is an all-one matrix with dimensions $f \times d$. In practice, $\lambda_H$ is first sampled from given $\mathcal{B}eta(\alpha_H, \alpha_H)$ distribution. Subsequently, we randomly select $\lfloor \lambda_H \times d \rfloor$ elements to construct the vector $s_h$ and the matrix $S_H$.

Since the embedding "dimensions" from different samples may be randomly combined in training, EXCELFORMER is encouraged to independently and equally handle various embedding dimensions. Considering each embedding dimension as a distinct "profile" version of input data (as each embedding element is projected from a scalar feature value), HID-MIX regularizes EXCELFORMER to behave like a bagging predictor (Breiman, 1996). Therefore, HID-MIX may also help mitigate the effects of data noise and perturbations, in addition to increasing the amount of training data.

**FEAT-MIX.** Our idea of FEAT-MIX is visualized as in Fig. 3. Unlike HID-MIX that operates on the embedding dimension, our FEAT-MIX synthesizes new sample $(x_m, y_m)$ by swapping parts of features between two randomly selected samples $x_1, x_2 \in \mathbb{R}^f$, and blending their labels $y_1$ and $y_2$ guided by feature importance, by:

$$\begin{cases} x_{\mathrm{m}} = \mathbf{s}_F \odot x_1 + (\mathbb{1}_F - \mathbf{s}_F) \odot x_2, \\ y_{\mathrm{m}} = \Lambda y_1 + (1 - \Lambda) y_2, \end{cases} \tag{8}$$

where the vector $\mathbf{s}_F$ and the all-one vector $\mathbb{1}_F$ are of size $f$, $\mathbf{s}_F$ contains $\lfloor \lambda_F \times f \rfloor$ randomly chosen elements set to 1 and the remaining elements set to 0. $\lambda_F \sim \mathcal{B}eta(\alpha_F, \alpha_F)$. The coefficient value, $\Lambda$, is determined based on the contribution of $x_1$ and $x_2$, taking into account feature importance, by:

$$\Lambda = \frac{\sum_{\mathbf{s}_F^{(i)} = 1} I(\mathbf{f}_i)}{\sum_{i=1}^{f} I(\mathbf{f}_i)}, \tag{9}$$

where $\mathbf{s}_F^{(i)}$ represents the $i$-th element of $\mathbf{s}_F$, and $I(\cdot)$ returns the feature importance using mutual information. When disregarding feature importance, $\Lambda = \lambda_F$ (assuming $\lfloor \lambda_F \times f \rfloor = \lambda_F \times f$), making FEAT-MIX degenerate into a form similar to cutmix (Yun et al., 2019). However, due to the presence of uninformative features in tabular datasets, FEAT-MIX emerges as a more robust scheme.

As features from two distinct samples are randomly combined to create new samples, FEAT-MIX promotes a solution with fewer feature interaction. This aligns with the functionality similar to our Interaction Attenuated Initialization (see Sec. 2.1). We argue that FEAT-MIX not only supplements the training dataset as a data augmentation method, but also encourages EXCELFORMER to predominantly exhibit characteristics of a non-rotationally invariant algorithm.

## 4 TRAINING METHODOLOGY AND LOSS FUNCTIONS

EXCELFORMER can handle both classification and regression tasks on tabular datasets in supervised learning. In training, our two proposed data augmentation schemes can be applied successively by HID-MIX(Embedding Layer(FEAT-MIX$(x, y)$)) or used independently. But, our tests suggest that the effect of EXCELFORMER on a certain dataset could be better by using only FEAT-MIX or HID-MIX. Thus, we use only one scheme in dealing with certain tabular datasets. The cross-entropy loss is used for classification tasks, and the mean square error loss is for regression tasks.

Table 1: Performance evaluation across 96 small-scale tabular datasets containing fewer than 10k samples. Each model underwent 5 independent trials, with the model's average rank ($\pm$ std) reported. The best ranks are highlighted in **bold** while the runners-up are underlined. Our EXCELFORMER consistently outperforms prior methods that undergo hyperparameter fine-tuning, regardless of whether EXCELFORMER uses fine-tuned or default hyperparameters. "d": using default hyperparameters; "t": using tuned hyperparameters; "No DA": neither FEAT-MIX nor HID-MIX is used.

| EXCELFORMER setting: | No DA (t) | FEAT-MIX (d) | HID-MIX (d) | Mix Tuned | Fully Tuned |
|---|---|---|---|---|---|
| XGboost (t) | 4.20±2.76 | 4.21±2.70 | 4.29±2.73 | 4.34±2.73 | 4.28±2.77 |
| Catboost (t) | 4.61±2.73 | 4.57±2.69 | 4.63±2.68 | 4.66±2.61 | 4.64±2.68 |
| FTT (t) | 4.32±2.36 | 4.35±2.35 | 4.41±2.25 | 4.44±2.32 | 4.39±2.37 |
| MLP (t) | 5.23±2.31 | 5.27±2.34 | 5.26±2.32 | 5.30±2.37 | 5.32±2.33 |
| DCN v2 (t) | 6.01±2.78 | 5.96±2.75 | 5.99±2.27 | 6.03±2.74 | 6.02±2.73 |
| AutoInt (t) | 5.70±2.61 | 5.78±2.51 | 5.77±2.56 | 5.88±2.53 | 5.80±2.55 |
| SAINT (t) | 5.48±2.59 | 5.48±2.55 | 5.56±2.56 | 5.61±2.55 | 5.56±2.58 |
| TransTab (d) | 6.78±2.52 | 6.80±2.59 | 6.82±2.57 | 6.86±2.59 | 6.87±2.55 |
| XTab (d) | 8.56±2.20 | 8.68±2.19 | 8.67±2.19 | 8.67±2.19 | 8.71±2.14 |
| EXCELFORMER (ours) | **4.11**±2.68 | **3.91**±2.60 | **3.62**±2.59 | **3.20**±2.10 | **3.41**±2.12 |

## 5 EXPERIMENTS

### 5.1 EXPERIMENTAL SETUPS

**Questions to Explore.** In this section, we will evaluate and inspect the property of EXCELFORMER and answer the following critical questions: (1) How does EXCELFORMER perform on small-scale datasets? (2) Does EXCELFORMER exhibit preferences for dataset types? (3) How do the key components in EXCELFORMER framework perform? (4) How does EXCELFORMER perform on larger datasets? (5) We also explore the non-rotationally invariance property in Appendix B.

**Implementation Details.** We configure the number of SPA and GRU modules as $L = 3$, set the feature embedding size to $d = 256$, and apply a dropout rate of 0.3 to the attention map. AdamW optimizer (Loshchilov & Hutter, 2018) is used with default settings. The learning rate is set to $10^{-4}$ without weight decay, and $\alpha_H$ and $\alpha_F$ for $\mathcal{B}eta$ distributions are both set to 0.5. These settings are the default hyperparameters for our EXCELFORMER. In the hyperparameter fine-tuning process, we utilized the Optuna library (Akiba et al., 2019) for all approaches. Consistent with (Gorishniy et al., 2021), we randomly select 80% of the data as training samples and the remaining 20% as test samples. During training, we reserve 20% training samples for validation. To fine-tune our EXCELFORMER, we designate two tuning configurations: "Mix Tuned" and "Fully Tuned". "Mix Tuned" refers to only fine-tune hyperparameters of data augmentation (for FEAT-MIX and HID-MIX), while "Fully Tuned" optimizes all hyperparameters, including those related to data augmentation and model architecture. A comprehensive description of all settings can be found in Appendix E. We applied early stopping with a patience of 32 for EXCELFORMER.

**Datasets.** A total of 96 small-scale datasets were employed, all of which were sourced from the Taptap dataset benchmark[2] based on the criterion of having a sample size less than 10,000. We also excluded multi-class classification datasets due to their limited quantity and susceptibility to evaluation biases stemming from label imbalance. We further evaluate EXCELFORMER on 21 larger public tabular datasets, ranging in scale from over 10,000 to 581,835 samples. The detailed dataset descriptions are provided in Appendix G.

**Compared Models.** We compare our new EXCELFORMER with two prominent GBDT approaches XGboost (Chen & Guestrin, 2016) and Catboost (Prokhorenkova et al., 2018) and several representative DNNs: FT-Transformer (FTT) (Gorishniy et al., 2021), SAINT (Somepalli et al., 2021), Multilayer Perceptron (MLP), DCN v2 (Wang et al., 2021a), AutoInt (Song et al., 2019), and TapPFN (Hollmann et al., 2022). We also include two pre-trained DNNs: TransTab (Wang & Sun, 2022) and XTab (Zhu et al., 2023) for reference. The implementations of XGboost and Catboost mainly follow (Gorishniy et al., 2021). Since we aim to extensively tune XGboost and Catboost for their best performances, we increase the number of estimators/ iterations (i.e., the number of decision trees) from 2000 to 4096 and the number of tuning iterations from 100 to 500, which give a

---

[2]`https://huggingface.co/datasets/ztphs980/taptap_datasets`

Table 2: Performance evaluation within several dataset subgroups. Performance rank within the datasets are reported. we also present the average normalized AUC and the average normalized $R^2$ for classification and regression tasks within the parentheses, respectively. The best scores are in **bold** and the runners-up are underlined. "(d)": default hyperparameters; "(t)": finely tuned hyperparameters. On GBDT-best datasets, we only mark the top two top-performing deep learning models, as it is unfair to compare them directly with GBDTs under this selection criterion.

| Model | EXCELFORMER | FTT (t) | XGb (t) | Cat (t) | MLP (t) | DCNv2 (t) | AutoInt (t) | SAINT (t) | TransTab (d) | XTab (d) | TabPFN (t) |
|---|---|---|---|---|---|---|---|---|---|---|---|
| Characteristics: Task Type | Classification [Rank (Ave Normalized AUC)] | | | | | | | | | | |
| Proportion | 51% | | | | | | | | | | |
| Setting: HID-MIX (d) | **3.88 (0.79)** | 4.88 (0.74) | 5.97 (0.63) | 5.77 (0.65) | 6.61 (0.60) | 6.38 (0.57) | 6.63 (0.56) | 6.07 (0.64) | 6.31 (0.64) | 9.50 (0.24) | 4.01 (0.78) |
| Setting: Mix Tuned | **3.78 (0.80)** | 4.91 (0.73) | 5.95 (0.62) | 5.79 (0.64) | 6.60 (0.59) | 6.39 (0.56) | 6.71 (0.56) | 6.10 (0.63) | 6.37 (0.63) | 9.46 (0.24) | 3.95 (0.78) |
| Characteristics: Task Type | Regression [Rank (Ave Normalized RMSE)] | | | | | | | | | | |
| Proportion | 49% | | | | | | | | | | |
| Setting: HID-MIX (d) | 3.81 (0.81) | 4.45 (0.78) | **3.43 (0.83)** | 4.26 (0.80) | 4.64 (0.74) | 6.26 (0.52) | 5.53 (0.65) | 5.64 (0.66) | 8.21 (0.33) | 8.79 (0.20) | / |
| Setting: Mix Tuned | 3.17 (0.86) | 4.49 (0.78) | 3.53 (0.82) | 4.28 (0.81) | 4.74 (0.74) | 6.32 (0.52) | 5.68 (0.65) | 5.72 (0.65) | 8.23 (0.33) | 8.83 (0.19) | / |
| Characteristics: #. Sample | ≥ 500 [Rank] | | | | | | | | | | |
| Proportion | 43% | | | | | | | | | | |
| Setting: HID-MIX (d) | **3.85** | 4.50 | 4.38 | 5.17 | 5.57 | 5.91 | 5.59 | 5.24 | 6.44 | 8.34 | / |
| Setting: Mix Tuned | **3.52** | 4.50 | 4.39 | 5.15 | 5.60 | 5.99 | 5.71 | 5.33 | 6.48 | 8.34 | / |
| Characteristics: #. Sample | < 500 [Rank] | | | | | | | | | | |
| Proportion | 57% | | | | | | | | | | |
| Setting: HID-MIX (d) | **3.45** | 4.34 | 4.22 | 4.23 | 5.02 | 6.05 | 5.90 | 5.79 | 7.10 | 8.92 | / |
| Setting: Mix Tuned | **3.18** | 4.38 | 4.28 | 4.29 | 5.05 | 6.05 | 5.97 | 5.79 | 7.13 | 8.88 | / |
| Characteristics: #. Feature | #. Feature < 8 [Rank] | | | | | | | | | | |
| Proportion | 32% | | | | | | | | | | |
| Setting: HID-MIX (d) | **3.45** | 3.84 | 3.98 | 5.08 | 4.23 | 6.32 | 6.16 | 5.32 | 7.52 | 9.10 | / |
| Setting: Mix Tuned | **3.27** | 3.84 | 4.03 | 5.06 | 4.34 | 6.26 | 6.21 | 5.35 | 7.50 | 9.13 | / |
| Characteristics: #. Feature | 8 ≤ #. Feature < 16 [Rank] | | | | | | | | | | |
| Proportion | 38% | | | | | | | | | | |
| Setting: HID-MIX (d) | **3.76** | 4.26 | 4.44 | 4.61 | 6.31 | 5.75 | 5.39 | 5.69 | 6.61 | 8.17 | / |
| Setting: Mix Tuned | **3.17** | 4.33 | 4.49 | 4.74 | 6.33 | 5.81 | 5.58 | 5.78 | 6.64 | 8.14 | / |
| Characteristics: #. Feature | #. Feature ≥ 16 [Rank] | | | | | | | | | | |
| Proportion | 30% | | | | | | | | | | |
| Setting: HID-MIX (d) | **3.62** | 5.19 | 4.41 | 4.17 | 5.05 | 5.93 | 5.81 | 5.64 | 6.33 | 8.84 | / |
| Setting: Mix Tuned | **3.17** | 5.22 | 4.48 | 4.14 | 5.05 | 6.05 | 5.90 | 5.69 | 6.45 | 8.84 | / |
| Characteristics: GBDT Performance | GBDT-best datasets in Classification Tasks [Rank (Ave Normalized AUC)] | | | | | | | | | | |
| Proportion | 31% | | | | | | | | | | |
| Setting: HID-MIX (d) | **4.63 (0.82)** | 4.70 (0.79) | 3.50 (0.84) | 3.73 (0.83) | 6.50 (0.65) | 6.70 (0.64) | 7.27 (0.53) | 7.43 (0.63) | 6.27 (0.69) | 10.03 (0.16) | 5.23 (0.71) |
| Setting: Mix Tuned | **3.28 (0.88)** | 4.63 (0.77) | 3.94 (0.82) | 3.81 (0.81) | 6.69 (0.64) | 6.97 (0.63) | 7.66 (0.53) | 7.41 (0.63) | 6.44 (0.68) | 10.09 (0.15) | 5.09 (0.72) |
| Characteristics: GBDT Performance | GBDT-best datasets in Regression Tasks [Rank (Ave Normalized RMSE)] | | | | | | | | | | |
| Proportion | 47% | | | | | | | | | | |
| Setting: HID-MIX (d) | **4.66 (0.70)** | 4.95 (0.70) | 1.95 (0.89) | 3.11 (0.85) | 5.41 (0.64) | 6.50 (0.47) | 5.86 (0.58) | 6.00 (0.55) | 8.14 (0.28) | 8.41 (0.21) | |
| Setting: Mix Tuned | **3.18 (0.82)** | 5.09 (0.69) | 2.18 (0.87) | 3.23 (0.85) | 5.55 (0.63) | 6.64 (0.47) | 6.18 (0.57) | 6.18 (0.54) | 8.27 (0.27) | 8.50 (0.18) | / |

more stringent setting and better performances. The settings for XGboost and Catboost are given in Appendix E. We use the default hyperparameters of pretrained models, TransTab and XTab, and fine-tune them on each dataset. They are not hyperparameter tuned, since their hyperparameter tuning spaces are not given. For large-scale datasets, FT-Transformer, SAINT, and TapFPN were fine-tuned based on the hyperparameters outlined in their respective papers. The architectures and hyperparameter tuning settings of the remaining DNNs follows the paper (Gorishniy et al., 2021). On small-scale datasets, we tuned 50 iterations for each datasets.

**Evaluation metrics.** For binary classification tasks, we compute the area under the ROC Curve (AUC) for evaluation. We use accuracy (ACC) for multi-class classification tasks. In regression tasks, we employ the negative root mean square error (nRMSE), where the negative sign is introduced to RMSE, aligning its direction with AUC and ACC, such that higher values across all these metrics indicate superior performance. Due to the high diversity among tabular datasets, performance ranks are used as a comprehensive metric, and the detailed results are given in Appendix H. To aggregate the results across datasets, we calculate the average normalized accuracy for multi-class classification tasks, average normalized AUC for binary classification tasks and average normalized nRMSE scores for regression tasks. The computational formula is provided in Appendix F.

## 5.2 RESULTS AND DISCUSSIONS

**Performances on Small-Scale Datasets.** To answer question (1), as depicted in Table 1, our EX-CELFORMER consistently outperforms other models that undergo dataset-adaptive hyperparameter tuning, regardless of whether the hyperparameters of the EXCELFORMER are tuned or not, which underscores the superiority of our proposed EXCELFORMER. We observe that EXCELFORMER with HID-MIX slightly outperforms that with FEAT-MIX; and if we tune hyperparameters of EX-CELFORMER, its performance achieves further improvement. Notably, hyperparameter fine-tuning reduces the standard deviations of performance ranks, indicating that applying hyperparameter fine-tuning onto EXCELFORMER can yield more consistently superior results. Interestingly, while fine-tuning all the hyperparameters ("Fully Tuned") should result in better performance ideally, it shows that, under the same fine-tuning iterations, "Mix Tuned" configuration performs better. This might be attributed to the higher efficiency of finely tuning data augmentation setting. To assess the effectiveness of our EXCELFORMER's architecture, we conducted experiments by excluding data augmentation (FEAT-MIX and HID-MIX) and compare it with existing works. The results show

Table 3: Performance evaluation across 21 larger-scale datasets, each containing more than 10,000 samples, is conducted. We report the average ranks along with their corresponding standard deviations, which are calculated based on the results of 5 runs with different random seeds. EXC. defines EXCELFORMER. The best and second best performances are **bold** and underlined.

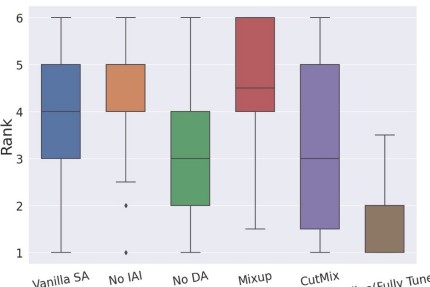

Figure 4: Ablation study. 'Vanilla SA': vanilla self-attention replacing SPA; 'No IAI': no interaction-attenuated initialization; 'No DA': no HID-MIX or FEAT-MIX; 'Mixup'/'CutMix': employing Mixup/CutMix for data augmentation.

| Setting | Model | Rank (mean ± std) |
|---|---|---|
| default hyperparameters | XGboost | 8.52 ± 1.86 |
| | Catboost | 7.52 ± 2.44 |
| | FTT | 6.71 ± 1.74 |
| | EXC. w/ FEAT-MIX | 6.62 ± 2.44 |
| | EXC. w/ HID-MIX | **4.76** ± 1.95 |
| hyperparameter fine-tuned | XGboost | 4.29 ± 2.59 |
| | Catboost | 6.24 ± 2.39 |
| | FT-T | 5.19 ± 2.60 |
| | EXC. (Mix Tuned) | 2.38 ± 1.53 |
| | EXC. (Fully Tuned) | **2.05** ± 1.40 |

that even without the use of FEAT-MIX and HID-MIX, EXCELFORMER still outperforms previous approaches, underscoring the superiority of our architectural design.

**Model Preference Inspection.** To answer question (2), we divide datasets into various subgroups according to the task type, dataset size, and the number of features, so as to examine models' performance within each subgroup. We adopt two configurations, HID-MIX (default) and Mix Tuned, for EXCELFORMER, while all of the prior works undergo hyperparameter fine-tuning. As shown in Table 2, EXCELFORMER with HID-MIX (default) exhibits the best performance in all subgroups except for regression tasks, where it slightly lags behind XGboost. The Mix Tuned EXCELFORMER significantly outperforms other models in all subgroups, indicating that EXCELFORMER does not exhibit overt dataset type preferences. Apart from EXCELFORMER, runner-up positions are occupied by TapFPN, FTT, XGboost, and Catboost. This suggests that, EXCELFORMER can achieve considerable performances in the majority of scenarios.

To further investigate the effectiveness of our EXCELFORMER on GBDT-best datasets, we selected datasets where XGBoost or CatBoost performed the best, excluding Excelformer. We observe that on these datasets, our model outperforms all previous deep learning approaches. Moreover, on GBDT-best classification datasets, Mix Tuned EXCELFORMER surpasses all both XGboost and Catboost. On GBDT-best regression datasets, our EXCELFORMER achieves competitive results comparable to CatBoost. On those datasets, EXCELFORMER even wins GBDTs on 11 out of 15 GBDT-best classification datasets and wins on 8 out of 22 GBDT-best regression datasets. It is crucial to note that the comparison on GBDT-best datasets is unfair to our EXCELFORMER, and it only serves as an observation of how our EXCELFORMER has altered the landscape of competition between GBDTs and deep learning models. In fact, the superiority of our EXCELFORMER (refer to Table 1 and Table 3) clearly demonstrates that EXCELFORMER has changed the situation of GBDTs' dominance on tabular data prediction tasks.

**Effect Inspection of Key Components.** To answer question (3), we conducted ablation studies on the key components of our model, namely, *semi-permeable attention*, interaction-attenuated initialization, and the data augmentation approach. We also compared the performance with the versions using vanilla Mixup (Zhang et al., 2018). As illustrated in Fig. 4, if the key components were excluded or replaced with counterparts, we observed varying degrees of performance degradation. The results in Table 1 indicate that HID-MIX performs slightly better than FEAT-MIX, but we found that FEAT-MIX and HID-MIX are better suited to different datasets. Notably, the performance of Mixup was worse than our data augmentation approach, and even worse than using no data augmentation. This suggests the conflicts we discussed in Sec. 3 between Mixup and the irregularities of feature-target functions. More detailed ablation studies are given in Appendix A.

**Performances on Larger Datasets.** To answer question (4), we conduct a comparison between our model and three previous state-of-the-art models: XGboost, Catboost, and FTT. We excluded other models from the comparison due to their relatively inferior performances and the significant computational load when dealing with large-scale datasets. Each model undergoes evaluation with two settings: using default hyperparameters and dataset-adaptive fine-tuning hyperparameters. As depicted in Table 3, strikingly, our model outperforms the previous models under both settings,

though it is not specifically designed for large datasets. This improvement is likely attributed to our method's enhanced data utilization. It is worth noting that our EXCELFORMER with HID-MIX still achieves comparable performance to prior models that undergo hyperparameter tuning, consistent with the findings on small-scale datasets. Different from the finding on small datasets, the Fully Tuned EXCELFORMER outperforms the Mix Tuned version on large datasets.

## 6 RELATED WORKS

**Supervised Tabular Data Prediction.** While deep neural networks (DNNs) have proven to be effective in computer vision (Khan et al., 2022) and natural language processing (Vaswani et al., 2017), GBDT approaches like XGBoost continue to be the preferred choice for tabular data prediction tasks (Katzir et al., 2020; Grinsztajn et al., 2022), particularly on smaller-scale datasets, due to their consistently superior performance. To enhance the performance of DNNs, recent studies have focused on developing sophisticated neural modules for (i) handling heterogeneous feature interactions (Gorishniy et al., 2021; Chen et al., 2022; Yan et al., 2023; Chen et al., 2023), (ii) seeking for decision paths by emulating decision-tree-like approaches (Katzir et al., 2020; Popov et al., 2019; Arik & Pfister, 2021), or (iii) resorting to conventional approaches (Cheng et al., 2016; Guo et al., 2017). In addition to model designs, various feature representation approaches, such as feature embedding (Gorishniy et al., 2022; Chen et al., 2023), discretization of continuous features (Guo et al., 2021; Wang et al., 2020), and Boolean algebra based methods (Wang et al., 2021b), were well explored. All these efforts suggested the potentials of DNNs, but they have not yet surpassed GBDTs in performance, especially on small-scale datasets. Moreover, there were several attempts (Wang & Sun, 2022; Arik & Pfister, 2021; Yoon et al., 2020; Zhu et al., 2023) to apply self-supervision learning on tabular datasets. However, many of these approaches are dataset- or domain-specific, and transferring these models to distant domains remains challenging due to the heterogeneity across tabular datasets. While pretrained on a substantial dataset corpus, XTap (Zhu et al., 2023) offered only a modest performance improvement due to the limited shared knowledge across datasets. TapPFN (Hollmann et al., 2022) concentrated on solving classification problems for small-scale tabular datasets and achieved commendable results. However, its efficiency waned when applied to larger datasets and regression tasks. In summary, compared to decision tree-based models, DNNs still fall short on tabular data, especially on small-scale ones, which remains an open challenge.

**Mixup and its Variants as Data Augmentation.** The vanilla Mixup (Zhang et al., 2018) generates a new data through convex interpolations of two existing data, which was proved beneficial on computer vision tasks (Tajbakhsh et al., 2020; Touvron et al., 2021a). However, we have observed that vanilla Mixup may conflict with irregular target patterns (please refer to Fig. 2 and Fig. 4) and typically achieves inferior performance. ManifoldMix (Verma et al., 2019) applied convex interpolations in the hidden states, which did not fundamentally alter the data synthesis approach of Mixup and exhibited similar characteristics to the vanilla Mixup. The follow-up variants CutMix (Yun et al., 2019), AttentiveMix (Walawalkar et al., 2020), SaliencyMix (Uddin et al., 2020), ResizeMix (Qin et al., 2020), and PuzzleMix (Kim et al., 2020b) spliced image pieces spatially, preserving local image patterns but being not directly applicable to tabular data. Kadra et al. (2021) explored various data augmentation approaches to improve the performance of MLP on tabular data. However, these data augmentation methods do not consistently apply effectively to various tabular datasets, necessitating time-consuming hyperparameter fine-tuning. In contrast, this paper introduced two novel Mixup-like data augmentation approaches for tabular data, HID-MIX and FEAT-MIX, which avoid the conflicts encountered with Mixup and contribute to EXCELFORMER achieving superior performance.

## 7 CONCLUSIONS

In this paper, we developed a new neural network called EXCELFORMER, accompanied by two data augmentation approaches HID-MIX and FEAT-MIX, for supervised prediction on small-scale tabular datasets. The key component of EXCELFORMER is the *semi-permeable attention* module (SPA), coupled with a unique interaction-attenuation initialization. This attention approach serves to reduce the influence of uninformative features and disrupt the rotational invariance property, thereby enhancing data utilization efficiency. Concurrently, HID-MIX and FEAT-MIX efficiently generate new samples, collaborating with SPA to enhance the performance of EXCELFORMER. Our proposed EXCELFORMER demonstrates superior performance compared to prior model, not only on small-scale datasets but on larger-scale datasets as well.

**Reproducibility.** 1) For a fair comparison, we used the publicly available Taptap tabular dataset benchmark (as stated in Sec. 5.1) to assess our model. We conducted 5 runs with different random seeds and reported the average results to mitigate random fluctuations, and std are also reported. We presented overall scores of model (including performances rank, average normalized AUC, average normalized nRMSE) in the main paper for comprehensive evaluation, and also provided detailed results in Appendices for reference. 2) The hyperparameter tuning spaces for our model and compared models are given in Sec. 5.1 and Appendix E. For compared methods, hyperparameter fine-tuning settings primarily adhered to the specifications outlined in the original papers or previous works. Any difference, such as increasing the number of estimators/trees for XGBoost and CatBoost to achieve better results, are explicitly noted. 3) Our code is attached in the supplementary material.

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

## A    ADDICTIVE EXAMINATION OF PROPOSED COMPONENTS

**Additive Study for Key Components.**    As a supplement to the ablation study in Sec. 5.2, we conduct an additive study to evaluate our proposed components on a vanilla Transformer. Specifically, we present the performances of a vanilla Transformer (i) with interaction-attenuated initialization (IAI), (ii) with semi-permeable attention (SPA), (iii) with both IAI and SPA, and EXCELFORMER (without any data augmentation). We respectively compute the performances on classification and regression tasks within the small-scale datasets, and the results are shown in Table 4. It is evident that all components make positive contributions to tabular data prediction tasks.

Table 4: Additive study for the effects of interaction-attenuated initialization (IAI) and semi-permeable attention (SPA), and the rest parts on 96 small-scale tabular datasets. "Class": binary classification; "Reg.": regression. No data augmentation is used on EXCELFORMER.

| | Transformer | Transformer + IAI | Transformer + SPA | Transformer + SPA + IAI | Excelformer |
|---|---|---|---|---|---|
| Class (Ave Norm AUC) | 0.232±0.36 | 0.683±0.37 | 0.289±0.37 | 0.753±0.33 | 0.775±0.30 |
| Reg. (Ave Norm nRMSE) | 0.138±0.30 | 0.738±0.25 | 0.751±0.26 | 0.766±0.35 | 0.911±0.16 |

**Effects of FEAT-MIX and HID-MIX.**    To verify the effects of FEAT-MIX and HID-MIX, we compare them with the feature resampling augmentation approach (Rubachev et al., 2022) on a simple MLP implemented following (Gorishniy et al., 2021). We respectively compute the performances on classification and regression tasks within the small-scale datasets. See Table 5, our HID-MIX outperforms the feature resampling approach, while the resampling approach performs better than FEAT-MIX. However, please note the significant variance (as indicated by the standard deviations), suggesting that each data augmentation approaches excel on specific datasets.

Table 5: Comparison of performance: FEAT-MIX, Feature Resampling, and HID-MIX on a simple MLP backbone across 96 small-scale datasets. "Class": binary classification; "Reg.": regression.

| backbone: MLP | feature resample | FEAT-MIX | HID-MIX | no data augmentation |
|---|---|---|---|---|
| Class (Ave Norm AUC) | 0.576±0.42 | 0.530±0.41 | 0.614±0.41 | 0.436±0.42 |
| Reg. (Ave Norm nRMSE) | 0.661±0.39 | 0.629±0.40 | 0.681±0.39 | 0.334±0.42 |
| Class (Ave Norm AUC) | 0.576±0.42 | 0.530±0.41 | 0.614±0.41 | 0.436±0.42 |
| Reg. (Ave Norm nRMSE) | 0.661±0.39 | 0.629±0.40 | 0.681±0.39 | 0.334±0.42 |

**Comparison between FEAT-MIX and CutMix.**    The primary distinction between the FEAT-MIX and CutMix approaches Yun et al. (2019) lies in whether the feature importance is considered when synthesizing new samples. To explore this difference, we conducted experiments on several datasets using the architecture of the EXCELFORMER as backbone. Our observations were made on both the original tables and the tables augmented with additional columns containing Gaussian noise. See Table 6, generally, FEAT-MIX outperforms CutMix or performs on par with CutMix on these datasets. However, in tables with noisy columns, we only observed a slight decline in the effectiveness of FEAT-MIX (with an improvement on the cpu dataset), while CutMix exhibited a more significant performance drop under the influence of noisy columns. Given the prevalence of numerous uninformative features in tabular data (Ng, 2004; Grinsztajn et al., 2022), the comparison of their performance and performance drops with noisy data emphasizes the importance of considering feature importance during interpolation. We find that FEAT-MIX stands out as a more resilient choice for tabular datasets.

## B    ROTATIONALLY INVARIANCE EVALUATION

Here we would like to inspect if the EXCELFORMER is more non-rotationally invariant and is more noise insensitive. We posit that the subpar performance of DNNs on small-scale datasets (as discussed in Sec. 1) can be partially attributed to their rotational invariance property, as well as the noise sensitivity caused by rotational invariance. Thus, we propose the *semi-permeable*

Table 6: Performance comparison between CutMix and FEAT-MIX. The first three datasets are for binary classification, with performance evaluated using the AUC (↑). The rest datasets are for regression, assessed through nRMSE (↑)."with noise" indicates we add some noisy columns to the table. Breast: Breast Cancer Coimbra; Diabetes: Pima-Indians-Diabetes; Campus:Campus Recruitment; yacht: yacht_hydrodynamics.

|  | Breast | Diabetes | Campus | cpu | fruitfly | yacht |
|---|---|---|---|---|---|---|
| CutMix | 0.702 | 0.822 | 0.972 | -102.06 | -16.19 | -3.59 |
| FEAT-MIX | **0.713** | **0.837** | **0.980** | **-79.10** | **-15.86** | **-0.83** |
| CutMix (with noise) | 0.688 | 0.809 | 0.938 | -115.10 | -17.09 | -4.40 |
| FEAT-MIX (with noise) | **0.700** | **0.834** | **0.969** | **-74.56** | **-16.60** | **-0.89** |
| Δ CutMix (↓) | 0.014 | 0.013 | 0.034 | 13.04 | 0.90 | 0.81 |
| Δ FEAT-MIX (↓) | 0.013 | 0.003 | 0.011 | -4.54 | 0.74 | 0.06 |

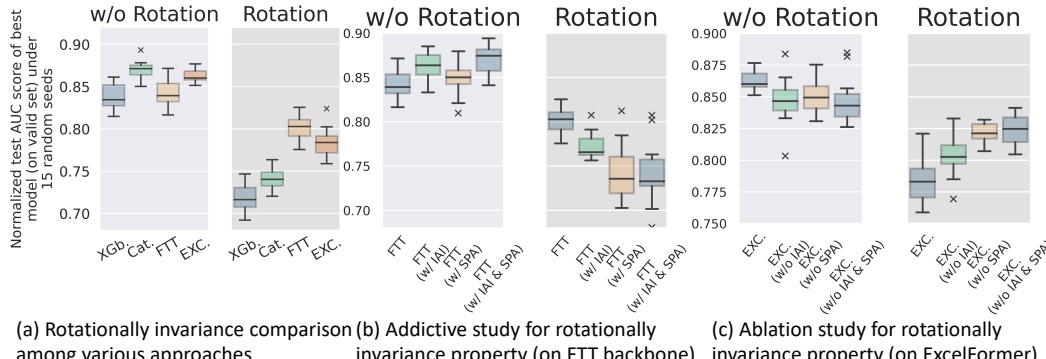

(a) Rotationally invariance comparison among various approaches
(b) Addictive study for rotationally invariance property (on FTT backbone)
(c) Ablation study for rotationally invariance property (on ExcelFormer)

Figure 5: Model performances under random dataset rotations. Test accuracy scores have been normalized across datasets, and the boxes represent the distribution of scores across 20 random seeds. XGb.: XGboost, Cat.: Catboost, EXC.: EXCELFORMER without data augmentation.

*attention*(SPA) with the interaction attenuated initialization (IAI) approach. Here we investigate whether these designs resolve those issues. We assess the test performance of EXCELFORMER (without using data augmentation) when randomly rotating the datasets. We utilize all classification datasets consisting of numerical features and containing fewer than 300 data samples. Additionally, we introduce $f$ uninformative features into each dataset (assuming that the original table comprises $f$ features), which are generated using Gaussian noises. As depicted in Fig. 5 (a), it is evident that after randomly rotating the datasets, XGBoost and CatBoost exhibit the most significant decline in performances. This observation suggests that they are algorithms with a higher degree of non-rotational invariance, aligning with the findings of (Ng, 2004). While the decline in performance of EXCELFORMER and FTT are not as substantial as those of decision tree-based models, it is still noticeable that EXCELFORMER's performance decreases by a larger extent after random rotations, compared to FTT. This observation indicates that our EXCELFORMER exhibits a higher degree of non-rotational variance compared to counterpart FTT.

Moreover, we conducted an additive study, utilizing FTT as the backbone and incorporating our proposed SPA and IAI on FTT. See Fig. 5(b), we find that: (i) both SPA and IAI contribute positively to the performances of FTT. (ii) In the presence of random dataset rotations, FTT with IAI and SPA demonstrated a more pronounced performance drop, thereby showcasing the efficacy of SPA and IAI in enhancing the non-rotational invariance property of FTT. Additionally, see Fig. 5(c), ablation studies on the EXCELFORMER backbone (where neither FEAT-MIX nor HID-MIX was applied) also highlighted the value of SPA and IAI in mitigating the rotational invariance property of DNN models.

Table 7: Performance of EXCELFORMER with other state-of-the-art models on additional 16 public multi-class classification datasets. "(t)": hyperparameter fine-tuning is performed. The best is marked in **bold** and the second best is underlined.

| Datasets | XGboost (t) | Catboost (t) | FTT(t) | MLP(t) | DCNv2(t) | AutoInt(t) | SAINT(t) | EXCELFORMER (t) |
|---|---|---|---|---|---|---|---|---|
| baseball | 0.948 | 0.940 | 0.918 | 0.922 | 0.892 | 0.910 | 0.929 | 0.951 |
| UCI-student-performance-mat | 0.506 | 0.405 | 0.380 | 0.114 | 0.177 | 0.278 | 0.329 | 0.582 |
| CPMP-2015-classification | 0.481 | 0.519 | 0.415 | 0.491 | 0.434 | 0.453 | 0.472 | 0.500 |
| arsenic-male-lung | 0.743 | 0.789 | 0.725 | 0.798 | 0.798 | 0.798 | 0.798 | 0.798 |
| braziltourism | 0.747 | 0.795 | 0.771 | 0.771 | 0.771 | 0.771 | 0.771 | 0.795 |
| segment | 0.961 | 0.961 | 0.957 | 0.961 | 0.961 | 0.970 | 0.961 | 0.981 |
| iris | 0.833 | 0.967 | 0.867 | 0.667 | 0.900 | 0.933 | 0.867 | 0.967 |
| analcatdata_broadwaymult | 0.439 | 0.596 | 0.474 | 0.439 | 0.439 | 0.456 | 0.491 | 0.526 |
| ipums_la_97-small | 0.482 | 0.484 | 0.474 | 0.447 | 0.465 | 0.476 | 0.452 | 0.491 |
| allrep | 0.981 | 0.983 | 0.964 | 0.967 | 0.964 | 0.966 | 0.967 | 0.984 |
| analcatdata_germangss | 0.438 | 0.388 | 0.463 | 0.375 | 0.363 | 0.450 | 0.425 | 0.475 |
| MyIris | 0.933 | 0.967 | 0.933 | 0.700 | 0.933 | 0.900 | 0.933 | 0.933 |
| JuanFeldmanIris | 0.933 | 0.933 | 0.900 | 0.800 | 0.900 | 0.933 | 0.867 | 0.933 |
| arrhythmia | 0.778 | 0.744 | 0.656 | 0.644 | 0.656 | 0.667 | 0.656 | 0.722 |
| wine-quality-red | 0.653 | 0.672 | 0.556 | 0.588 | 0.594 | 0.594 | 0.547 | 0.597 |
| glass | 0.791 | 0.767 | 0.698 | 0.674 | 0.581 | 0.628 | 0.628 | 0.651 |
| rank ($\pm$std) | 3.91$\pm$2.28 | 2.53$\pm$1.70 | 5.53$\pm$1.85 | 6.09$\pm$1.88 | 5.97$\pm$1.61 | 4.66$\pm$1.65 | 5.22$\pm$1.41 | **2.09$\pm$1.23** |
| average normalized ACC ($\pm$std) | 0.65$\pm$0.35 | 0.82$\pm$0.27 | 0.39$\pm$0.33 | 0.25$\pm$0.31 | 0.33$\pm$0.35 | 0.50$\pm$0.31 | 0.42$\pm$0.29 | **0.85$\pm$0.24** |

## C  ADDITIONAL MULTI-CLASS CLASSIFICATION RESULTS

To further assess the effectiveness of EXCELFORMER in multi-class classification, we present the performance results of EXCELFORMER alongside other models across an additional set of 16 multi-class classification datasets, as detailed in Table 7. The corresponding performance ranks and average normalized accuracy are also provided. It is evident from the results that our EXCELFORMER also outperforms the compared approaches in various multi-class classification tasks.

## D  AVERAGE NORMALIZED SCORES ON LARGE-SCALE DATASETS

Table 8: Performance evaluation on larger-scale datasets, each containing more than 10,000 samples. Average normalized scores of accuracy (for multi-class classification datasets), AUC (for binary-class classification datasets), and nRMSE (for regression datasets) are used.

| Setting | Model | binclass | regression | multiclass |
|---|---|---|---|---|
| default hyperparameter | XGboost | 0 $\pm$ 0 | 0.470$\pm$0.400 | 0.218$\pm$ 0.400 |
| | Catboost | 0.977$\pm$0.034 | 0.286$\pm$0.291 | 0.280$\pm$0.408 |
| | FTT | 0.807$\pm$0.397 | 0.613$\pm$0.313 | 0.763$\pm$0.163 |
| | EXCELFORMER w/ FEAT-MIX | 0.982$\pm$0.022 | 0.513$\pm$0.422 | 0.791$\pm$0.143 |
| | EXCELFORMER w/ HID-MIX | **0.976**$\pm$0.030 | **0.825**$\pm$0.279 | **0.990**$\pm$0.020 |
| hyperparameter fine-tuned | XGboost | 0.409$\pm$0.424 | 0.756$\pm$0.285 | 0.258$\pm$0.241 |
| | Catboost | 0.281$\pm$0.365 | 0.276$\pm$0.385 | 0.095$\pm$0.135 |
| | FT-T | 0.316$\pm$0.387 | 0.428$\pm$0.398 | 0.596$\pm$0.425 |
| | EXCELFORMER (Mix Tuned) | 0.526$\pm$0.413 | **0.865**$\pm$0.150 | 0.735$\pm$0.286 |
| | EXCELFORMER (Fully Tuned) | **0.777**$\pm$0.393 | **0.865**$\pm$0.203 | **0.957**$\pm$0.085 |

## E  DETAILS OF HYPER-PARAMETER FINE-TUNING SETTINGS

For XGboost and Catboost, we follow the implementations and settings in (Gorishniy et al., 2021), while increasing the number of estimators/iterations (i.e., decision trees) and the number of tuning iterations, so as to attain better performance. For our EXCELFORMER, we apply the Optuna based tuning (Akiba et al., 2019). The hyper-parameter search spaces of EXCELFORMER, XGboost, and Catboost are reported in Table 9, Table 10, and Table 11, respectively. For EXCELFORMER, we tune just 50 iterations on the configurations with regard to the data augmentation (it is marked as "Mix Tuned"). For "Fully Tuned" version, we finely tune 50 interations on all the hyper-parameters.

Table 9: The hyper-parameter optimization space for EXCELFORMER. The items marked with "*" are used to obtain a "Mix Tuned" EXCELFORMER, while all the items are used to obtain a "Fully Tuned" version.

| Hyper-parameter | Distribution |
|---|---|
| #. Layers $L$ | UniformInt[2, 5] |
| Representation size $d$ | {64, 128, 256} |
| #. Heads | {4, 8, 16, 32} |
| Residual dropout rate | Uniform[0, 0.5] |
| Learning rate | LogUniform[$3 \times 10^{-5}, 10^{-3}$] |
| Weight decay | {0.0, LogUniform[$10^{-6}, 10^{-3}$]} |
| (*) Mixup type | {FEAT-MIX, HID-MIX, neither} |
| (*) $\alpha$ of $\mathcal{B}eta$ distribution | Uniform[0.1, 3.0] |

Table 10: The hyper-parameter tuning space for XGboost.

| Hyper-parameter | Distribution |
|---|---|
| Booster | "gbtree" |
| N-estimators | Const(4096) |
| Early-stopping-rounds | Const(50) |
| Max depth | UniformInt[3, 10] |
| Min child weight | LogUniform[$10^{-8}, 10^5$] |
| Subsample | Uniform[0.5, 1.0] |
| Learning rate | LogUniform[$10^{-5}, 1$] |
| Col sample by level | Uniform[0.5, 1] |
| Col sample by tree | Uniform[0.5, 1] |
| Gamma | {0, LogUniform[$10^{-8}, 10^2$]} |
| Lambda | {0, LogUniform[$10^{-8}, 10^2$]} |
| Alpha | {0, LogUniform[$10^{-8}, 10^2$]} |
| #. Tuning iterations | 500 |

Table 11: The hyper-parameter tuning space for Catboost.

| Hyper-parameter | Distribution |
|---|---|
| Iterations (number of trees) | Const(4096) |
| Od pval | Const(0.001) |
| Early-stopping-rounds | Const(50) |
| Max depth | UniformInt[3, 10] |
| Learning rate | LogUniform[$10^{-5}, 1$] |
| Bagging temperature | Uniform[0, 1] |
| L2 leaf reg | LogUniform[1, 10] |
| Leaf estimation iterations | UniformInt[1, 10] |
| #. Tuning iterations | 500 |

## F  IMPLEMENTATION DETAILS OF METRICS USED IN THIS WORK

**Feature Importance.**   In this study, we employ Normalized Mutual Information (NMI) to assess the importance of various features, as mutual information can capture dependencies between features and targets. We implement NMI using the sklearn Python package. Specifically, for classification tasks, we utilize the "feature_selection.mutual_info_classif" function, and for regression tasks, we utilize the "feature_selection.mutual_info_regression" function.

**Average Normalized Scores across Datasets.**   To aggregate the model performances across datasets, we calculate the average normalized scores (Wistuba et al., 2015) for AUC, accuracy, and nRMSE to comprehensively evaluate the model performances. Specifically, we first normalize the scores among the compared models for given datasets, and then average them across datasets. Formally, among $D$ involved datasets, the average normalized score $s_m$ for the model $m$ is computed by:

$$s'_{m,d} = \frac{s_{m,d} - \min_{i \in M_0}(s_{i,d})}{\max_{i \in M_0}(s_{i,d}) - \min_{i \in M_0}(s_{i,d})}, s_m = \frac{\sum_{d=1}^{D} s'_{m,d}}{D} \tag{10}$$

where $M_0$ encompasses all the models compared. The $s_m$ can be specified as $\text{AUC}_m$, $\text{ACC}_m$, or $\text{nRMSE}_m$. We calculate AUC, ACC, and nRMSE separately because these metrics correspond to different tasks and exhibit varying sensitivities to errors.

**Performance Rank.**   We performed 5 runs with different random seeds and calculated the average results for each dataset. Additionally, we computed the overall rank across datasets for comparison. Average rank is given to tied values.

# G  DETAILED DESCRIPTION OF DATASETS USED

The details of the 96 used small-scale tabular datasets are summarized in Table 12 and Table 13. The details of the 21 large-scale datasets are summarized in Table 14. We use the same train-valid-test split for all the approaches.

# H  DETAILED RESULTS ON SMALL- AND LARGE- SCALE DATASETS

We present the average results (five runs averaged) of all the models for each dataset. The results for the 96 small-scale datasets can be found in Table 15, and the performance on the 21 large-scale datasets is provided in Table 16.

Table 12: The details of the 96 small-scale tabular datasets used. "#. Num" and "#. Cat" denote the numbers of numerical and categorical features, respectively. "#. Sample" presents the size of a dataset.

| Dataset | #. Sample | #. Feature | #. Num | #.Cat | Task Type |
|---|---|---|---|---|---|
| Analytics Vidhya Loan Prediction | 614 | 11 | 5 | 6 | binclass |
| Audit Data | 776 | 24 | 21 | 3 | binclass |
| Automobiles | 201 | 25 | 13 | 12 | binclass |
| Bigg Boss India | 567 | 21 | 6 | 15 | binclass |
| Breast Cancer Dataset | 569 | 30 | 30 | 0 | binclass |
| Campus Recruitment | 215 | 13 | 6 | 7 | binclass |
| chronic kidney disease | 400 | 13 | 9 | 4 | binclass |
| House Price | 506 | 17 | 14 | 3 | binclass |
| Compositions of Glass | 214 | 9 | 9 | 0 | binclass |
| Credit Card Approval | 590 | 15 | 6 | 9 | binclass |
| Customer Classification | 1000 | 11 | 5 | 6 | binclass |
| Development Index | 225 | 6 | 6 | 0 | binclass |
| fitbit dataset | 457 | 13 | 12 | 1 | binclass |
| Horse Colic Dataset | 299 | 27 | 9 | 18 | binclass |
| Penguins Classified | 344 | 6 | 4 | 2 | binclass |
| Pima-Indians_Diabetes | 768 | 8 | 8 | 0 | binclass |
| Real Estate DataSet | 511 | 13 | 11 | 2 | binclass |
| Startup Success Prediction | 923 | 45 | 9 | 36 | binclass |
| Store Data Performance | 135 | 16 | 7 | 9 | binclass |
| The Estonia Disaster Passenger List | 989 | 6 | 1 | 5 | binclass |
| AAPL_stock_price_2021_2022 | 346 | 5 | 5 | 0 | regression |
| AAPL_stock_price_2021_2022_1 | 347 | 5 | 5 | 0 | regression |
| AAPL_stock_price_2021_2022_2 | 348 | 5 | 5 | 0 | regression |
| analcatdata_creditscore | 100 | 6 | 3 | 3 | binclass |
| analcatdata_homerun | 162 | 26 | 12 | 14 | regression |
| analcatdata_lawsuit | 264 | 4 | 3 | 1 | binclass |
| analcatdata_vineyard | 468 | 3 | 1 | 2 | regression |
| auto_price | 159 | 15 | 13 | 2 | regression |
| autoPrice | 159 | 15 | 14 | 1 | regression |
| bodyfat | 252 | 14 | 14 | 0 | regression |
| boston | 506 | 13 | 11 | 2 | regression |
| boston_corrected | 506 | 19 | 15 | 4 | regression |
| Boston-house-price-data | 506 | 13 | 11 | 2 | regression |
| cholesterol | 303 | 13 | 7 | 6 | regression |
| cleveland | 303 | 13 | 7 | 6 | regression |
| cloud | 108 | 5 | 3 | 2 | regression |
| cps_85_wages | 534 | 10 | 3 | 7 | regression |
| cpu | 209 | 7 | 5 | 2 | regression |
| DEE | 365 | 6 | 6 | 0 | regression |
| Diabetes-Data-Set | 768 | 8 | 8 | 0 | binclass |
| DiabeticMellitus | 281 | 97 | 6 | 91 | binclass |
| disclosure_x_bias | 662 | 3 | 3 | 0 | regression |
| disclosure_x_noise | 662 | 3 | 3 | 0 | regression |
| disclosure_x_tampered | 662 | 3 | 3 | 0 | regression |
| disclosure_z | 662 | 3 | 3 | 0 | regression |
| echoMonths | 130 | 9 | 7 | 2 | regression |
| EgyptianSkulls | 150 | 4 | 3 | 1 | regression |
| ELE-1 | 495 | 2 | 2 | 0 | regression |
| fishcatch | 158 | 7 | 5 | 2 | regression |
| Fish-market | 159 | 6 | 5 | 1 | regression |

Table 13: The details of the 96 small-scale tabular datasets used (continued). "#. Num" and "#. Cat" denote the numbers of numerical and categorical features, respectively. "#. Sample" presents the size of a dataset.

| Dataset | #. Sample | #. Feature | #. Num | #.Cat | Task Type |
|---|---|---|---|---|---|
| forest_fires | 517 | 12 | 8 | 4 | regression |
| Forest-Fire-Area | 517 | 12 | 8 | 4 | regression |
| fruitfly | 125 | 4 | 2 | 2 | regression |
| HappinessRank_2015 | 158 | 9 | 8 | 1 | regression |
| Heart_disease_classification | 296 | 13 | 7 | 6 | binclass |
| hungarian | 294 | 13 | 11 | 2 | binclass |
| Indian-Liver-Patient-Patient | 583 | 11 | 9 | 2 | binclass |
| Intersectional-Bias-Assessment | 1000 | 18 | 14 | 4 | binclass |
| liver-disorders | 345 | 5 | 5 | 0 | regression |
| lowbwt | 189 | 9 | 2 | 7 | regression |
| lungcancer_shedden | 442 | 23 | 20 | 3 | regression |
| machine_cpu | 209 | 6 | 6 | 0 | regression |
| meta | 528 | 21 | 16 | 5 | regression |
| nki70.arff | 144 | 76 | 72 | 4 | binclass |
| no2 | 500 | 7 | 7 | 0 | regression |
| pharynx | 195 | 10 | 3 | 7 | regression |
| Pima-Indians-Diabetes | 768 | 8 | 8 | 0 | binclass |
| pm10 | 500 | 7 | 7 | 0 | regression |
| Pokmon-Legendary-Data | 801 | 12 | 9 | 3 | binclass |
| Reading_Hydro | 1000 | 26 | 11 | 15 | regression |
| residential_building | 372 | 108 | 100 | 8 | regression |
| rmftsa_ladata | 508 | 10 | 10 | 0 | regression |
| strikes | 625 | 6 | 6 | 0 | regression |
| student-grade-pass-or-fail-prediction | 395 | 29 | 4 | 25 | binclass |
| Swiss-banknote-conterfeit-detection | 200 | 6 | 6 | 0 | binclass |
| The-Estonia-Disaster-Passenger-List | 989 | 6 | 1 | 5 | binclass |
| The-Office-Dataset | 188 | 10 | 2 | 8 | regression |
| tokyo1 | 959 | 44 | 42 | 2 | binclass |
| visualizing_environmental | 111 | 3 | 3 | 0 | regression |
| weather_ankara | 321 | 9 | 9 | 0 | regression |
| wisconsin | 194 | 32 | 32 | 0 | regression |
| yacht_hydrodynamics | 308 | 6 | 6 | 0 | regression |
| Absenteeism at work | 740 | 20 | 7 | 13 | binclass |
| Audit Data | 776 | 24 | 21 | 3 | binclass |
| Breast Cancer Coimbra | 116 | 9 | 9 | 0 | binclass |
| Cervical cancer (Risk Factors) | 858 | 30 | 25 | 5 | binclass |
| Climate Model Simulation Crashes | 540 | 19 | 18 | 1 | binclass |
| Early stage diabetes risk prediction | 520 | 16 | 1 | 15 | binclass |
| extention of Z-Alizadeh sani dataset | 303 | 57 | 20 | 37 | binclass |
| HCV data | 615 | 12 | 11 | 1 | binclass |
| Heart failure clinical records | 299 | 12 | 7 | 5 | binclass |
| Parkinson Dataset | 240 | 46 | 44 | 2 | binclass |
| QSAR Bioconcentration classes | 779 | 11 | 7 | 4 | binclass |
| Quality Assessment of DC | 97 | 62 | 62 | 0 | binclass |
| User Knowledge Modeling | 258 | 5 | 5 | 0 | binclass |
| Z-Alizadeh Sani | 303 | 54 | 20 | 34 | binclass |

Table 14: The details of 21 large-scale datasets used. "#. Num" and "#. Cat" denote the numbers of numerical and categorical features, respectively. "#. Sample" presents the size of a dataset.

| Dataset | Abbr. | Task Type | #. Features | #. Num | #. Cat | #. Sample | Link |
|---|---|---|---|---|---|---|---|
| sulfur | SU | regression | 6 | 6 | 0 | 10,081 | https://www.openml.org/d/44145 |
| bank-marketing | BA | binclass | 7 | 7 | 0 | 10,578 | https://www.openml.org/d/44126 |
| Brazilian_houses | BR | regression | 8 | 8 | 0 | 10,692 | https://www.openml.org/d/44141 |
| eye | EY | multiclass | 26 | 26 | 0 | 10,936 | http://www.cis.hut.fi/eyechallenge2005 |
| MagicTelescope | MA | binclass | 10 | 10 | 0 | 13,376 | https://www.openml.org/d/44125 |
| Ailerons | AI | regression | 33 | 33 | 0 | 13,750 | https://www.openml.org/d/44137 |
| pol | PO | regression | 26 | 26 | 0 | 15,000 | https://www.openml.org/d/722 |
| binarized-pol | BP | binclass | 48 | 48 | 0 | 15,000 | https://www.openml.org/d/722 |
| credit | CR | binclass | 10 | 10 | 0 | 16,714 | https://www.openml.org/d/44089 |
| california | CA | regression | 8 | 8 | 0 | 20,640 | https://www.dcc.fc.up.pt/~ltorgo/Regression/cal_housing.html |
| house_sales | HS | regression | 15 | 15 | 0 | 21,613 | https://www.openml.org/d/44144 |
| house | HO | regression | 16 | 16 | 0 | 22,784 | https://www.openml.org/d/574 |
| diamonds | DI | regression | 6 | 6 | 0 | 53,940 | https://www.openml.org/d/44140 |
| helena | HE | multiclass | 27 | 27 | 0 | 65,196 | https://www.openml.org/d/41169 |
| jannis | JA | multiclass | 54 | 54 | 0 | 83,733 | https://www.openml.org/d/41168 |
| higgs-small | HI | binclass | 28 | 28 | 0 | 98,049 | https://www.openml.org/d/23512 |
| road-safety | RO | binclass | 32 | 29 | 3 | 111,762 | https://www.openml.org/d/44161 |
| medicalcharges | ME | regression | 3 | 3 | 0 | 163,065 | https://www.openml.org/d/44146 |
| SGEMM_GPU_kernel_performance | SG | regression | 9 | 3 | 6 | 241,600 | https://www.openml.org/d/44069 |
| covtype | CO | multiclass | 54 | 54 | 0 | 581,012 | https://www.openml.org/d/1596 |
| nyc-taxi-green-dec-2016 | NY | regression | 9 | 9 | 0 | 581,835 | https://www.openml.org/d/44143 |

Table 15: Performance of EXCELFORMER with other state-of-the-art models on 96 public small-scale datasets. E.: EXCELFORMER; E. + F: EXCELFORMER with FEAT-MIX; E. + H: EXCELFORMER with HID-MIX; XGb: XGboost, Cat: Catboost; FTT: FT-Transformer; TapP: TabPFN; TT: TransTab; XT: XTab. "(d)": using default hyperparameters; "(t)": hyperparameter fine-tuning is performed. "(M)": Mix Tuned version; "(F)": Fully Tuned version. TabPFN is designed for classification, we mark "n/a" in regression tasks.

| Datasets | E.+F (d) | E.+H (d) | E. (MT) | E.(FT) | FTT (t) | XGb (t) | Cat (t) | MLP (t) | DCNv2 (t) | AutoInt(t) | SAINT (t) | TT (d) | XT(d) | TapP(t) |
|---|---|---|---|---|---|---|---|---|---|---|---|---|---|---|
| Analytics Vidhya Loan Prediction | 0.7449 | 0.7421 | 0.7421 | 0.7421 | 0.7285 | 0.7486 | 0.7045 | 0.7359 | 0.7464 | 0.7350 | 0.7505 | 0.7291 | 0.7240 | 0.7331 |
| Audit Data | 0.9984 | 0.9905 | 0.9991 | 0.9941 | 0.9993 | 0.9995 | 1.0000 | 0.9936 | 0.9990 | 0.9983 | 0.9995 | 0.9983 | 0.9822 | 0.9998 |
| Automobiles | 0.9774 | 0.9798 | 0.9869 | 0.9750 | 0.9583 | 0.9679 | 0.9726 | 0.9512 | 0.9226 | 0.9726 | 0.9536 | 0.9679 | 0.9545 | 0.9845 |
| Bigg Boss India | 1.0000 | 1.0000 | 0.9861 | 0.9861 | 0.9799 | 1.0000 | 0.9861 | 0.9954 | 0.9985 | 0.9059 | 0.9753 | 1.0000 | 0.9769 | 1.0000 |
| Breast Cancer Dataset | 0.9970 | 0.9937 | 0.9921 | 0.9944 | 0.9841 | 0.9914 | 0.9851 | 0.9917 | 0.9828 | 0.9795 | 0.9828 | 0.9970 | 0.9927 | 0.9916 |
| Campus Recruitment | 0.9795 | 0.9846 | 0.9795 | 0.9487 | 0.9487 | 0.9795 | 0.9667 | 0.9256 | 0.9103 | 0.9744 | 0.9590 | 0.9436 | 0.9232 | 0.9821 |
| chronic kidney disease | 0.9993 | 1.0000 | 0.9967 | 0.9960 | 0.9960 | 0.9900 | 0.9940 | 0.9907 | 0.9767 | 0.9893 | 0.9853 | 0.9947 | 0.9973 | 0.9993 |
| House Price | 0.8904 | 0.9015 | 0.9015 | 0.8983 | 0.9011 | 0.8818 | 0.8977 | 0.9026 | 0.8924 | 0.9013 | 0.9054 | 0.8971 | 0.8863 | 0.9007 |
| Compositions of Glass | 0.8976 | 0.8595 | 0.8976 | 0.9238 | 0.8595 | 0.8655 | 0.9024 | 0.8167 | 0.9000 | 0.6929 | 0.7929 | 0.7929 | 0.7976 | 0.8905 |
| Credit Card Approval | 0.9583 | 0.9719 | 0.9680 | 0.9701 | 0.9607 | 0.9595 | 0.9680 | 0.9447 | 0.9478 | 0.9550 | 0.9550 | 0.9662 | 0.9376 | 0.9553 |
| Customer Classification | 0.5792 | 0.5719 | 0.5280 | 0.6014 | 0.5951 | 0.4727 | 0.5802 | 0.5376 | 0.6417 | 0.5952 | 0.6348 | 0.6125 | 0.6134 | 0.6229 |
| Development Index | 1.0000 | 0.9671 | 1.0000 | 1.0000 | 0.9815 | 0.9259 | 0.9805 | 0.9588 | 0.9362 | 0.9633 | 1.0000 | 0.9856 | 0.9210 | 0.9856 |
| fitbit dataset | 0.7991 | 0.8216 | 0.8092 | 0.8025 | 0.8154 | 0.8138 | 0.7975 | 0.8073 | 0.8101 | 0.7628 | 0.7939 | 0.8164 | 0.7847 | 0.8914 |
| Horse Colic Dataset | 0.7523 | 0.7477 | 0.7373 | 0.7407 | 0.6921 | 0.6921 | 0.7269 | 0.7025 | 0.6169 | 0.6038 | 0.7083 | 0.7199 | 0.6905 | 0.7346 |
| Penguins Classified | 0.9991 | 0.9991 | 0.9991 | 1.0000 | 1.0000 | 0.9966 | 0.9944 | 0.9983 | 0.9983 | 0.9954 | 1.0000 | 0.9940 | 0.9872 | 0.9983 |
| Pima-Indians_Diabetes | 0.8367 | 0.8330 | 0.8154 | 0.8148 | 0.8048 | 0.7887 | 0.7667 | 0.8222 | 0.8069 | 0.7987 | 0.8096 | 0.8135 | 0.7546 | 0.8181 |
| Real Estate DataSet | 0.9176 | 0.8894 | 0.9041 | 0.9176 | 0.9045 | 0.9010 | 0.9167 | 0.8621 | 0.8760 | 0.9014 | 0.8945 | 0.8883 | 0.8471 | 0.9029 |
| Startup Success Prediction | 0.9938 | 0.9914 | 0.8397 | 0.8445 | 0.8464 | 0.9937 | 0.7277 | 0.8373 | 0.7701 | 0.8456 | 0.8363 | 0.8204 | 0.8371 | 0.8428 |
| Store Data Performance | 0.7632 | 0.7632 | 0.6908 | 0.7566 | 0.6382 | 0.6316 | 0.7829 | 0.6447 | 0.6711 | 0.7627 | 0.6184 | 0.6645 | 0.6513 | 0.8487 |
| The Estonia Disaster Passenger List | 0.7572 | 0.7559 | 0.7708 | 0.7713 | 0.7546 | 0.7379 | 0.7379 | 0.7464 | 0.7401 | 0.7379 | 0.7436 | 0.7505 | 0.7248 | 0.7518 |
| analcatdata_creditscore | 1.0000 | 1.0000 | 1.0000 | 1.0000 | 1.0000 | 0.9400 | 0.9667 | 1.0000 | 0.9200 | 0.9667 | 0.8533 | 1.0000 | 0.8467 | 1.0000 |
| analcatdata_lawsuit | 1.0000 | 1.0000 | 1.0000 | 1.0000 | 1.0000 | 1.0000 | 1.0000 | 1.0000 | 0.9898 | 0.9541 | 0.9847 | 0.9796 | 0.9395 | 1.0000 |
| Diabetes-Data-Set | 0.8356 | 0.8337 | 0.8248 | 0.8294 | 0.8241 | 0.7852 | 0.7754 | 0.8215 | 0.7954 | 0.8311 | 0.8257 | 0.8120 | 0.7903 | 0.8152 |
| DiabeticMellitus | 0.9878 | 0.9865 | 0.9865 | 0.9865 | 0.9784 | 0.9743 | 0.9493 | 0.7405 | 0.8242 | 0.8932 | 0.9203 | 0.9041 | 0.7346 | 0.9405 |
| Heart_disease_classification | 0.8984 | 0.9096 | 0.9152 | 0.9241 | 0.9342 | 0.8990 | 0.9018 | 0.8984 | 0.9342 | 0.9241 | 0.9174 | 0.9107 | 0.9152 | 0.9252 |
| hungarian | 0.8446 | 0.8596 | 0.8596 | 0.8484 | 0.8822 | 0.8578 | 0.8910 | 0.9273 | 0.9060 | 0.8534 | 0.9261 | 0.8985 | 0.9273 | 0.8722 |
| Indian-Liver-Patient-Patient | 0.7133 | 0.6984 | 0.7197 | 0.7232 | 0.7551 | 0.7399 | 0.7206 | 0.7310 | 0.7133 | 0.7254 | 0.7332 | 0.7332 | 0.6935 | 0.7381 |
| Intersectional-Bias-Assessment | 0.9953 | 0.9958 | 0.9960 | 0.9979 | 0.9916 | 0.9925 | 0.9977 | 0.9956 | 0.9944 | 0.9962 | 0.9982 | 0.9942 | 0.9736 | 0.9965 |
| nki70.arff | 0.8158 | 0.9263 | 0.8867 | 0.8737 | 0.6263 | 0.8526 | 0.8526 | 0.6842 | 0.6684 | 0.8105 | 0.8353 | 0.8263 | 0.8263 | 0.9216 |
| Pima-Indians-Diabetes | 0.8356 | 0.8337 | 0.8109 | 0.8052 | 0.8494 | 0.8140 | 0.7528 | 0.7974 | 0.7931 | 0.7638 | 0.7806 | 0.8141 | 0.7903 | 0.8152 |
| Pokmon-Legendary-Data | 0.9767 | 0.9888 | 0.9981 | 0.9869 | 0.9679 | 0.9504 | 0.9840 | 0.9203 | 0.9412 | 0.9917 | 0.9772 | 0.9810 | 0.9679 | 0.9801 |
| student-grade-pass-or-fail-prediction | 0.9898 | 0.9884 | 1.0000 | 1.0000 | 0.9710 | 1.0000 | 1.0000 | 0.9771 | 0.8211 | 0.9993 | 0.9608 | 0.9898 | 0.8494 | 0.9601 |
| Swiss-banknote-conterfeit-detection | 0.9925 | 0.9950 | 1.0000 | 1.0000 | 0.9975 | 1.0000 | 1.0000 | 1.0000 | 0.9975 | 1.0000 | 1.0000 | 1.0000 | 0.9450 | 1.0000 |
| The-Estonia-Disaster-Passenger-List | 0.7572 | 0.7559 | 0.7555 | 0.7585 | 0.7719 | 0.7723 | 0.7366 | 0.7485 | 0.7530 | 0.7390 | 0.7433 | 0.7464 | 0.6948 | 0.7518 |
| tokyo1 | 0.9715 | 0.9740 | 0.9696 | 0.9702 | 0.9692 | 0.9706 | 0.9536 | 0.9716 | 0.9727 | 0.9665 | 0.9653 | 0.9595 | 0.9596 | 0.9705 |
| Absenteeism at work | 0.8579 | 0.8665 | 0.8669 | 0.8921 | 0.8281 | 0.8402 | 0.8145 | 0.8278 | 0.7973 | 0.8299 | 0.8119 | 0.8352 | 0.7713 | 0.8339 |
| Audit Data | 0.9984 | 0.9905 | 0.9995 | 0.9995 | 1.0000 | 1.0000 | 1.0000 | 0.9934 | 0.9967 | 0.9991 | 0.9995 | 0.9967 | 0.9822 | 0.9998 |
| Breast Cancer Coimbra | 0.7133 | 0.7622 | 0.9276 | 0.7483 | 0.7483 | 0.7762 | 0.7692 | 0.6993 | 0.7168 | 0.6503 | 0.6434 | 0.6923 | 0.6224 | 0.5923 |
| Cervical cancer (Risk Factors) | 0.7137 | 0.6431 | 0.6431 | 0.5415 | 0.6194 | 0.6039 | 0.7165 | 0.5680 | 0.5867 | 0.5720 | 0.4839 | 0.5268 | 0.5353 | 0.9798 |
| Climate Model Simulation Crashes | 0.9574 | 0.9776 | 0.9742 | 0.9776 | 0.9192 | 0.9529 | 0.9832 | 0.9439 | 0.9439 | 0.8563 | 0.9733 | 0.7957 | 0.7677 | 0.9973 |
| Early stage diabetes risk prediction | 0.9906 | 0.9684 | 0.9746 | 0.9820 | 0.9793 | 0.9969 | 0.9906 | 0.9957 | 0.9973 | 0.9984 | 0.9980 | 0.9922 | 0.9109 | 0.9629 |
| extention of Z-Alizadeh sani dataset | 0.9638 | 0.9651 | 0.9651 | 0.9121 | 0.9638 | 0.9606 | 0.9509 | 0.9522 | 0.9651 | 0.9457 | 0.9574 | 0.9134 | 0.8416 | 1.0000 |
| HCV data | 0.9982 | 0.9942 | 0.9965 | 0.9988 | 0.9924 | 0.9930 | 0.9714 | 1.0000 | 0.9982 | 1.0000 | 0.9977 | 1.0000 | 0.9523 | 0.8511 |
| Heart failure clinical records | 0.8652 | 0.8883 | 0.8806 | 0.8922 | 0.9127 | 0.8633 | 0.8177 | 0.8370 | 0.8395 | 0.8691 | 0.8203 | 0.8588 | 0.7918 | 0.9214 |
| Parkinson Dataset | 0.9167 | 0.9253 | 0.9392 | 0.9253 | 0.9306 | 0.8559 | 0.9071 | 0.9201 | 0.9253 | 0.9253 | 0.9207 | 0.9132 | 0.8941 | 0.9121 |
| QSAR Bioconcentration classes | 0.7721 | 0.8331 | 0.8314 | 0.8454 | 0.8796 | 0.8308 | 0.8363 | 0.8419 | 0.8640 | 0.8242 | 0.8162 | 0.8293 | 0.8079 | 0.8613 |
| Quality Assessment of DC | 0.5490 | 0.5294 | 0.5294 | 0.8235 | 0.3922 | 0.6078 | 0.9412 | 0.7451 | 0.7451 | 0.2941 | 0.1961 | 0.4314 | 0.3922 | 0.3137 |
| User Knowledge Modeling | 0.9771 | 0.9771 | 0.9771 | 0.9559 | 0.9902 | 0.9330 | 0.9673 | 0.9641 | 0.9559 | 0.9673 | 0.9739 | 0.8713 | 0.8547 | 0.9886 |
| Z-Alizadeh Sani | 0.8385 | 0.8863 | 0.8863 | 0.8618 | 0.8773 | 0.8424 | 0.8863 | 0.8618 | 0.8760 | 0.8450 | 0.8734 | 0.8592 | 0.8041 | 0.8527 |
| AAPL_stock_price_2021_2022 | -2.4201 | -0.8485 | -1.1187 | -0.6742 | -1.0613 | -0.4714 | -1.2488 | -0.3812 | -2.9911 | -1.1469 | -1.2134 | -3.7808 | -3.2483 | n/a |
| AAPL_stock_price_2021_2022_1 | -1.3351 | -0.7599 | -0.3584 | -0.3108 | -0.6781 | -0.7711 | -1.4369 | -0.7036 | -1.2721 | -1.0302 | -2.4377 | -2.7802 | -2.5602 | n/a |
| AAPL_stock_price_2021_2022_2 | -1.4472 | -0.5832 | -0.3367 | -0.3141 | -0.4005 | -0.7059 | -0.9954 | -0.2768 | -1.2930 | -0.9359 | -2.2318 | -2.8244 | -2.4182 | n/a |
| analcatdata_homerun | -0.7584 | -0.9188 | -0.7432 | -0.7514 | -0.7456 | -0.8075 | -0.7366 | -0.7425 | -0.7731 | -0.7706 | -0.7452 | -0.7574 | -0.8417 | n/a |
| analcatdata_vineyard | -2.9582 | -2.7122 | -3.0034 | -2.6116 | -2.4820 | -2.1594 | -2.3821 | -2.4602 | -2.4509 | -2.4403 | -3.5946 | -3.7126 | n/a |
| auto_price | -1751.0 | -2103.5 | -1830.7 | -2463.1 | -2244.8 | -1720.8 | -1935.9 | -2702.4 | -2503.7 | -2020.4 | -2905.1 | -3100.4 | -3031.4 | n/a |
| autoPrice | -1751.0 | -2103.5 | -1676.9 | -1831.3 | -2341.6 | -1659.6 | -1977.4 | -2623.2 | -4026.9 | -1840.2 | -3104.0 | -3093.8 | -2969.5 | n/a |
| bodyfat | -1.0621 | -0.7597 | -0.8297 | -0.5658 | -0.5431 | -0.8420 | -1.1247 | -1.6816 | -3.9424 | -1.7095 | -1.4271 | -4.0332 | -4.3167 | n/a |
| boston | -2.7724 | -2.9132 | -3.0481 | -3.2503 | -4.2412 | -3.0366 | -3.5042 | -3.6890 | -4.0360 | -4.2976 | -4.5132 | -4.9288 | -4.6731 | n/a |
| boston_corrected | -3.0906 | -3.6336 | -3.3379 | -3.3726 | -3.3748 | -3.2464 | -3.6352 | -3.4899 | -3.2328 | -3.3248 | -3.8338 | -5.5182 | -5.7764 | n/a |
| Boston-house-price-data | -3.1074 | -2.9132 | -3.0481 | -3.9133 | -3.4856 | -3.1320 | -3.5397 | -4.4732 | -5.6720 | -4.1353 | -3.9253 | -4.7881 | -4.6171 | n/a |
| cholesterol | -63.898 | -63.607 | -62.204 | -61.527 | -61.702 | -60.718 | -61.791 | -62.435 | -61.145 | -62.760 | -62.621 | -61.434 | -64.213 | n/a |
| cleveland | -0.8839 | -0.8765 | -0.8686 | -0.8853 | -0.9944 | -0.8863 | -0.8918 | -0.9546 | -0.9936 | -0.9455 | -0.8704 | -1.1198 | -1.2134 | n/a |
| cloud | -0.5701 | -0.6858 | -0.4539 | -0.6851 | -0.4608 | -0.2720 | -0.3458 | -0.6079 | -0.6326 | -0.8258 | -0.7637 | -0.9437 | -1.0297 | n/a |
| cps_85_wages | -4.3197 | -4.4261 | -4.2873 | -4.3968 | -4.2237 | -4.6278 | -4.6009 | -4.4573 | -4.2974 | -4.2999 | -4.4403 | -4.7683 | -4.8651 | n/a |
| cpu | -79.104 | -76.943 | -76.402 | -91.466 | -95.975 | -62.504 | -104.760 | -74.299 | -68.783 | -122.468 | -123.213 | -137.357 | -131.2979 | n/a |
| DEE | -0.4023 | -0.4255 | -0.4294 | -0.4278 | -0.3863 | -0.4051 | -0.4239 | -0.3974 | -0.8244 | -0.4174 | -0.4296 | -0.6657 | -0.4780 | n/a |
| disclosure_x_bias | -21921 | -21743 | -21919 | -21876 | -21807 | -22587 | -21853 | -22022 | -21878 | -21912 | -22159 | -22481 | -23453 | n/a |
| disclosure_x_noise | -26993 | -27266 | -26843 | -26919 | -27196 | -26943 | -27438 | -27044 | -27944 | -27232 | -27010 | -27412 | -27078 | n/a |
| disclosure_x_tampered | -27168 | -27275 | -27824 | -27062 | -27245 | -27318 | -27647 | -27180 | -27227 | -27984 | -27114 | -27347 | -27018 | n/a |
| disclosure_z | -21506 | -21374 | -21496 | -21477 | -21791 | -21911 | -21815 | -21753 | -30624 | -21764 | -21804 | -22272 | -23509 | n/a |
| echoMonths | -8.8668 | -9.4200 | -10.1428 | -9.9251 | -11.5086 | -12.6059 | -10.4651 | -11.1439 | -12.4521 | -10.1922 | -9.5287 | -13.5465 | -14.0546 | n/a |
| EgyptianSkulls | -1425.98 | -1393.52 | -1403.98 | -1403.98 | -1360.73 | -1460.25 | -1487.83 | -1519.55 | -1243.86 | -1480.12 | -1603.05 | -1575.59 | -1669.02 | n/a |
| ELE-1 | -736.04 | -736.25 | -758.72 | -749.40 | -749.72 | -770.96 | -782.94 | -761.12 | -739.36 | -779.54 | -737.94 | -838.68 | -816.96 | n/a |
| fishcatch | -50.863 | -46.911 | -86.628 | -76.929 | -66.500 | -79.260 | -102.405 | -126.658 | -89.525 | -155.355 | -149.660 | -180.285 | -205.606 | n/a |
| Fish-market | -72.073 | -70.419 | -76.484 | -80.037 | -88.557 | -63.291 | -70.877 | -112.847 | -64.847 | -128.376 | -96.060 | -153.163 | -172.934 | n/a |
| forest_fires | -109.375 | -109.339 | -108.763 | -107.853 | -109.139 | -108.803 | -107.700 | -108.925 | -108.707 | -108.573 | -109.064 | -108.921 | -108.578 | n/a |
| Forest-Fire-Area | -109.375 | -109.339 | -108.988 | -107.538 | -109.026 | -108.803 | -106.091 | -108.945 | -109.292 | -109.428 | -109.349 | -109.015 | -108.578 | n/a |
| fruitfly | -15.856 | -15.752 | -15.858 | -15.732 | -16.438 | -20.724 | -16.224 | -16.251 | -18.620 | -17.561 | -16.023 | -15.829 | -21.850 | n/a |
| HappinessRank_2015 | -0.1402 | -0.0753 | -0.0640 | -0.0753 | -0.0800 | -0.0856 | -0.1765 | -0.2066 | -0.1220 | -0.3450 | -0.2596 | -0.9244 | -1.2549 | n/a |
| liver-disorders | -3.1001 | -2.9445 | -2.9602 | -3.0291 | -3.2481 | -3.0613 | -2.9170 | -2.9404 | -3.3161 | -3.0200 | -2.8904 | -3.2965 | n/a |
| lowbwt | -419.24 | -421.87 | -417.13 | -419.41 | -451.40 | -422.76 | -443.85 | -447.74 | -486.07 | -406.45 | -420.31 | -501.78 | -580.11 | n/a |
| lungcancer_shedden | -2.8148 | -2.5672 | -2.6234 | -2.6200 | -2.7049 | -2.7345 | -2.5833 | -2.5232 | -2.5298 | -2.8645 | -2.5481 | -2.8069 | -3.4472 | n/a |
| machine_cpu | -71.259 | -85.958 | -90.238 | -82.287 | -92.617 | -78.152 | -107.735 | -73.420 | -89.281 | -125.633 | -129.315 | -187.953 | -177.951 | n/a |
| meta | -153.09 | -162.95 | -142.67 | -128.98 | -164.11 | -147.92 | -236.52 | -141.32 | -142.79 | -237.33 | -146.03 | -192.56 | -273.34 | n/a |
| no2 | -0.5015 | -0.4864 | -0.4972 | -0.4967 | -0.4948 | -0.4912 | -0.5082 | -0.5289 | -0.5212 | -0.5127 | -0.4985 | -0.6629 | -0.7692 | n/a |
| pharynx | -286.57 | -281.51 | -277.71 | -273.68 | -310.59 | -279.05 | -277.78 | -337.05 | -492.43 | -270.79 | -282.70 | -328.23 | -391.29 | n/a |
| pm10 | -0.7670 | -0.7650 | -0.7267 | -0.8022 | -0.8010 | -0.7487 | -0.7331 | -0.8141 | -0.9794 | -0.7942 | -0.8064 | -0.8130 | -0.9376 | n/a |
| Reading_Hydro | -0.0039 | -0.0037 | -0.0040 | -0.0039 | -0.0038 | -0.0036 | -0.0037 | -0.0039 | -0.0041 | -0.0042 | -0.0047 | -0.0188 | -0.0081 | n/a |
| residential_building | -351.46 | -210.01 | -168.25 | -196.44 | -237.14 | -200.64 | -306.75 | -533.02 | -723.89 | -533.37 | -584.36 | -643.16 | n/a |
| rmftsa_ladata | -2.0287 | -1.8550 | -1.8238 | -2.0475 | -2.0305 | -2.0150 | -1.8144 | -2.0999 | -2.4473 | -2.3571 | -2.0216 | -2.5843 | -2.5447 | n/a |
| strikes | -586.03 | -592.41 | -587.24 | -604.66 | -604.16 | -592.34 | -588.58 | -620.24 | -660.56 | -589.25 | -599.57 | -615.12 | -637.11 | n/a |
| The-Office-Dataset | -0.3876 | -0.4189 | -0.3654 | -0.4127 | -0.4152 | -0.3350 | -0.3730 | -0.4068 | -0.4148 | -0.4843 | -0.5396 | -0.5697 | n/a |
| visualizing_environmental | -2.5584 | -2.8069 | -3.0343 | -3.4924 | -2.8716 | -2.3128 | -2.8110 | -2.5520 | -2.9296 | -3.1022 | -2.7731 | -3.4982 | -3.8523 | n/a |
| weather_ankara | -1.7222 | -1.3999 | -1.5609 | -1.4824 | -1.5755 | -1.8900 | -1.5884 | -2.0655 | -2.4473 | -2.3743 | -3.3254 | -2.9359 | n/a |
| wisconsin | -36.915 | -37.548 | -38.315 | -38.603 | -36.128 | -35.500 | -35.720 | -34.429 | -75.613 | -34.541 | -37.677 | -37.229 | -51.180 | n/a |
| yacht_hydrodynamics | -0.8310 | -0.9270 | -0.7151 | -1.0738 | -1.0881 | -0.7432 | -1.0243 | -1.2074 | -1.2786 | -2.3096 | -4.4713 | -5.1386 | -6.5417 | n/a |

Table 16: Performance of EXCELFORMER with other state-of-the-art models on 21 public large-scale datasets. EXC.: EXCELFORMER; EXC. + F: EXCELFORMER with FEAT-MIX; EXC. + H: EX-CELFORMER with HID-MIX; XGb: XGboost, Cat: Catboost; "(d)": using default hyperparameters; "(t)": hyperparameter fine-tuning is performed. "(M)": Mix Tuned; "(F)": Fully Tuned.

| Datasets | XGb (d) | Cat (d) | FTT (d) | Exc. + F (d) | Exc. + H (d) | XGb (t) | Cat (t) | FTT (t) | Exc. (M) | Exc. (F) |
|---|---|---|---|---|---|---|---|---|---|---|
| SU | -0.02025 | -0.01994 | -0.01825 | -0.01840 | -0.01740 | -0.01770 | -0.02200 | -0.01920 | -0.01730 | -0.01610 |
| BA | 80.25 | 89.20 | 88.26 | 89.00 | 88.65 | 88.97 | 89.16 | 88.64 | 89.21 | 89.16 |
| BR | -0.07667 | -0.07655 | -0.07390 | -0.11230 | -0.06960 | -0.07690 | -0.09310 | -0.07940 | -0.06270 | -0.06410 |
| EY | 69.97 | 69.85 | 71.06 | 71.44 | 72.09 | 72.88 | 72.41 | 71.73 | 74.14 | 78.94 |
| MA | 86.21 | 93.83 | 93.66 | 93.38 | 93.66 | 93.69 | 93.66 | 93.69 | 94.04 | 94.11 |
| AI | -0.0001669 | -0.0001652 | -0.0001637 | -0.0001689 | -0.0001627 | -0.0001605 | -0.0001616 | -0.0001641 | -0.0001615 | -0.0001612 |
| PO | -5.342 | -6.495 | -4.675 | -5.694 | -2.862 | -4.331 | -4.622 | -2.705 | -2.629 | -2.636 |
| BP | 99.13 | 99.95 | 99.13 | 99.94 | 99.95 | 99.96 | 99.95 | 99.97 | 99.93 | 99.96 |
| CR | 76.55 | 85.15 | 85.22 | 85.23 | 85.22 | 85.11 | 85.12 | 85.19 | 85.26 | 85.36 |
| CA | -0.4707 | -0.4573 | -0.4657 | -0.4331 | -0.4587 | -0.4359 | -0.4359 | -0.4679 | -0.4316 | -0.4336 |
| HS | -0.1815 | -0.1790 | -0.1740 | -0.1835 | -0.1773 | -0.1707 | -0.1746 | -0.1734 | -0.1726 | -0.1727 |
| HO | -3.368 | -3.258 | -3.208 | -3.305 | -3.147 | -3.139 | -3.279 | -3.142 | -3.159 | -3.214 |
| DI | -0.2372 | -0.2395 | -0.2378 | -0.2368 | -0.2387 | -0.2353 | -0.2362 | -0.2389 | -0.2359 | -0.2358 |
| HE | 35.02 | 37.77 | 37.38 | 37.22 | 38.20 | 37.39 | 37.81 | 38.86 | 38.65 | 38.61 |
| JA | 71.62 | 71.92 | 72.67 | 72.51 | 72.79 | 72.45 | 71.97 | 73.15 | 73.15 | 73.55 |
| HI | 71.59 | 80.31 | 80.65 | 80.60 | 80.75 | 80.28 | 80.22 | 80.71 | 80.88 | 81.22 |
| RO | 80.42 | 87.98 | 88.51 | 88.65 | 88.15 | 90.48 | 89.55 | 89.29 | 89.33 | 89.27 |
| ME | -0.0819 | -0.0835 | -0.0845 | -0.0821 | -0.0808 | -0.0820 | -0.0829 | -0.0811 | -0.0809 | -0.0808 |
| SG | -0.01658 | -0.03377 | -0.01866 | -0.01587 | -0.01531 | -0.01635 | -0.02038 | -0.01644 | -0.01465 | -0.01454 |
| CO | 96.42 | 92.13 | 96.71 | 97.38 | 97.17 | 96.92 | 96.25 | 97.00 | 97.43 | 97.43 |
| NY | -0.3805 | -0.4459 | -0.4135 | -0.3887 | -0.3930 | -0.3683 | -0.3808 | -0.4248 | -0.3710 | -0.3625 |

