# OpenReview forum: "ExcelFormer: Making Neural Network Excel in Small Tabular Data Prediction"
_ICLR.cc/2024/Conference — Submitted to ICLR 2024_

### Official Review · Reviewer_Wsjx · 2023-10-15

**Soundness:** 2 fair
**Presentation:** 3 good
**Contribution:** 2 fair
**Rating:** 5
**Confidence:** 5

**Summary:**

**The paper studies** machine learning problems on small (<1K objects) tabular datasets (e.g. classification, regression, etc.) with additional experiments on larger (up to 500K objects) datasets.

**The main contribution of the paper** is ExcelFormer -- a deep learning scheme (Transformer-like architecture + custom training recipe) with the following new elements compared to the vanilla Transformer:
- (architecture) custom attention
- (architecture) custom feed-forward block
- (architecture) custom feature embeddings
- (architecture) custom prediction head
- (training) custom initialization
- (training) two custom augmentations

**The main claim:** *"EXCELFORMER consistently and substantially outperforms previous works"*

**Strengths:**

- The story is mostly easy to follow (also, I like the main illustration!).
- The research direction (designing better tabular deep learning architectures and augmentations) is important.
- The experimental part includes many datasets.
- I like the idea of using feature importances (1) to guide the attention between features and (2) to guide one of the two proposed data augmentations.

**Weaknesses:**

(1) (major) **Many *orthogonal* changes (listed in the summary above) are proposed *at once*.** It makes it difficuilt to attribute the observed results to any single element, which I believe to be important in the research context, especially for this genre of papers. I believe that the elements should be introduced either in isolation or step-by-step, but not at once (unfortunately, ablating each of the elements using the *final* architecture does not addresses the issue). Also, in my opinion, each of the elements should be compared against existing alternatives (i.e. the proposed augmentations VS existing augmentations, the proposed embeddings VS the existing embeddings, etc.).

Overall, modifying the well-established Transformer architecture in six(!) different aspects (listed in the summary), most of which has dedicated research subfields looks like an extremely ambitious goal to me. And I respect that, however, it makes it extremely hard to properly introduce and analyse each of the elements.

(2) (major) In my opinion, **the storyline around rotation invariance should be extended with specific analysis/experiments/results. Purely intuitive guidance may not be enough to drive the design decisions.** There are multiple places where the *formal* term "rotation invariance" is used in *informal* ways. For example, the paper uses terms like "more/nearly non-rotationally invariant". Overall, there is nothing wrong with relying on intuition, but after a certain threshold, there is a risk of coming to wrong conclusions.

A potential solution is to design a dedicated experiment that will quantify rotation invariance of any ML model. Then, some of the proposed elements can be motivated as a way to reduce the invariance according to the designed experiment. Again, this should be done *when introducing the elements*, not with the final architecture (as in Figure 5).

(3) Unfortunately, in my opinion, **the novelty is limited.** Some of the proposed modifications (listed in the summary above) are technically new, however, from the same technical perspective, they remain similar to the existing alternatives.

(4) In my opinion, sharing code, starting from the review stage, is important for this kind of studies. I wish I had an opportunity to have a look at the code to review the experimental setup and implementation details.

(5) Instead of Paragraph 2 of Section 1, I recommend writing only ~2-3 high level sentences and then referring to Section 5.4 of "Why do tree-based models still outperform deep learning on tabular data?" by Grinsztajn et al.

(6) I recommend proof-reading the paper for English style, vocabulary and grammar issues.

**Questions:**

How exactly is the mutual information computed for the continuous features and for regression labels?

---

> ### Author Response · Authors · 2023-11-22
> **To reviewer Wsjx (part 1)**
>
> Thank you for helpful comments. We trust that our responses will effectively address all your concerns.
>
> ---
>
> > **Q1**: Many *orthogonal* changes are proposed at once? Need more ablation studyies.
>
> **A1**: Thank you for your concerns. We would like to clarify it from 2 aspects:
>
> **(1) More detailed studies**. We have included more ablation studies and addictive studies to inspect the performances of proposed components.
>
> 1. Addictive study: on a vanilla Transformer (Tra), we show that **all of our proposed modules**, SPA, IAI, and GLU projection/embedding (GLU projection and embedding share the idea, and both of them process features through the GLU) **make positive contributions to tabular data prediction tasks**, and the contributions are cumulative.
>
> | | Tra | Tra + IAI | Tra + SPA | Tra + SPA + IAI |Tra + SPA + IAI + GLU|
> |-|-|-|-|-|-|
> | binclass (Ave Norm AUC, ↑) | 0.232±0.36 | 0.683±0.37| 0.289±0.37 | 0.753±0.33  | 0.775±0.30|
> | regression (Ave Norm nRMSE, ↑) | 0.138±0.30  | 0.738±0.25  | 0.751±0.26|0.766±0.35 | 0.911±0.16|
>
> 2. Compare our data augmentation with Mixup and CutMix. Ablation study in Figure 4 show CutMix and Mixup perform worse than our proposed data augmentation. To analyze the difference of cutmix and Feat-Mix carefully, in Table 6, we present the results of cutmix and Feat-Mix on several datasets. Theoretically, the difference between cutmix and feat-mix lies in the use of feature importance. Empirically, we find that:
>
> * In general, **Feat-Mix outperforms Cutmix** due to its ability to provide more accurate labels by considering feature importance.
>
> * When noise exists in the data, **Feat-Mix is less affected by the noise while cutmix is more heavily impacted**. Since tabular datasets have been shown to contain many uninformative features`[1]`, we believe Feat-Mix is a superior choice.
>
> The detailed results:
>
> | | Breast | Diabetes | Campus | cpu | fruitfly | yacht |
> |:-:|:-:|:-:|:-:|:-:|:-:|:-:|
> | CutMix (↑) | 0.702 | 0.822 | 0.972 | -102.06 | -16.19 | -3.59 |
> | Feat-Mix (↑) | **0.713** | **0.837** | **0.980** | **-79.10** | **-15.86** | **-0.83** |
> | CutMix (on noisy data) (↑) | 0.688 | 0.809 | 0.938 | -115.10 | -17.09 | -4.40 |
> | Feat-Mix (on noisy data) (↑) | **0.700** | **0.834** | **0.969** | **-74.56** | **-16.60** | **-0.89** |
> | $\Delta$ CutMix (↓) | 0.014 | 0.013 | 0.034 | 13.04 | 0.90 | 0.81 |
> | $\Delta$ Feat-Mix (↓) | **0.013** | **0.003** | **0.011** | **-4.54** | **0.74** | **0.06** |
>
> 3. Compare Hid-Mix, Feat-Mix with Feature resample (an augmentation for tabular data) on MLP:
>
> The results are presented below. We observe that Hid-Mix outperforms Feature Resample and all these approach dominate different parts of datasets.
>
> |   | Feature Resample | Feat-Mix | Hid-Mix | No data augmentation |
> |-|--|--|-|--|
> | Binclass (Ave Norm AUC, ↑) | 0.576±0.42 | 0.530±0.41 | 0.614±0.41 | 0.436±0.42 |
> | regression (Ave Norm nRMSE, ↑)|0.661±0.39|0.629±0.40 | 0.681±0.39 | 0.334±0.42 |
>
> 4. An ablation study is shown in Figure 4 that the performances of Excelformer drops when removing one of those components.
>
> **(2) Why propose those components?**
>
> 1. We would like to point out a specific issue of the tabular data prediction domain: **Tabular datasets are highly diverse and heterogeneous. Unlike other fields (such as computer vision) where a single model can dominate most datasets, different previous tabular data prediction models only excel on a part of datasets (see Table 2: excluding Excelformer, the best models vary in different groups). Excelformer is the first work that realizes such dominance.**
>
> 2. Given the diversity of datasets in this field and the no-free-lunch principle, finding a single component that is universally applicable to all types of datasets is challenging (previous works have faltered in this aspect). **In fact, the orthogonality of methods is indeed the cornerstone of our contribution. When addressing the diversity of tabular datasets and tasks, the proposal and integration of orthogonal methods augment the robustness and efficacy of Excelformer**. Then, we achieve:
>
> *  **Using default hyperparameters (it is very convenient in application!)**, ExcelFormer outperforms previous hyperparameter-tuned models;
>
> * ExcelFormer achieves robust performance **across various types of datasets / tasks** (see Table 2)
>
> **We argue that AI development aims to facilitate practical applications.** The combination of versatility and the convenience of not requiring hyperparameter tuning positions Excelformer as an excellent practical tool for tabular data prediction tasks, especially for users with limited computer science proficiency or computational resources. **Please note that no previous work attains performances as dominating as Excelformer, making this a key contribution.**
>
> ---
>
> References:
>
> `[1]` Why do tree-based models still outperform deep learning on typical tabular data? NeurIPS, 2022.

---

> ### Author Response · Authors · 2023-11-22
> **To reviewer Wsjx (part 2)**
>
> > **Q2**: Novelty?
>
> **A2**: Thank you for this concern. I believe the evaluation of novelty is quite objective. I suggest considering the changes an approach brings as a reference for novelty creation. Let's examine what changes we made by introducing Excelformer:
>
> * BEFORE EXCELFORMER:
>
>   * The overall performance of deep learning approaches is **inferior** to GBDTs.
>   * Different previous models excels in specific types of datasets / tasks (see Table 2).  It is **challenging to determine a "good" model** before conducting an evaluation.
>   * **Fine-tuning hyperparameters is crucial** for attaining state-of-the-art results across all previous models. This poses challenges for users with limited proficiency in computer science or access to computational resources.
>
> * AFTER EXCELFORMER IS PROPOSED:
>   * Excelformer (a DL approach) beats GBDTs and various DL competitors;
>   * Excelformer outperforms previous models on various types of datasets / tasks (see Table 2). It is easier for users to choose a model (Excelformer) now.
>    * Although fine-tuning hyperparameters is beneficial for Excelformer, using its default hyperparameters can achieve top results.
>
> * To achieve this, THE COST WE INCUR IS MINIMAL:
>   * Obviously, the computational efficiency of SPA is the same as vanilla self-attention module.
>   * Interaction Attenuated Initialization is almost computational free.
>   * As analyzed in Sec. 2.2, the computational complexity of GLU projection is equivalent to that of vanilla FFN.
>   * Feat-Mix and Hid-Mix are nearly computational free.
>
> We will highlight the contribution of Excelformer in the final version.
>
> Considering the changes that Excelformer has brought to the field of tabular data prediction, we would appreciate it if you could reconsider your evaluation of novelty. Thank you!
>
> ---
>
> > **Q3**: In my opinion, the storyline around rotation invariance should be extended with specific analysis/experiments/results.
>
> **A3**: Thank you for your constructive suggestion! In the initial version, we conducted an analysis of the rotation invariance property, i.e. the subfig (a) in Figure 5. Following your suggestion, we have taken a step further by providing an additive study and an ablation study to analyze the effects of SPA and IAI on breaking the rotation invariance of deep learning models.
>
> * Additive study setting: We use SPA and IAI on the backbone of FTT `[1]`
>
> * Ablation study setting: We exclude SPA and IAI from the architecture of Excelformer
>
> We find that:
>
> * With SPA and IAI, our Excelformer shows better non-rotationally invariance, compared with previous works.
>
> * On the FTT backbone, the addition of SPA and IAI makes positive contributions to break the rotational invariance property.
>
> * On the Excelformer architecture, removing SPA and IAI weakens Excelformer's rotational invariance property.
>
> * **The effects of adding or removing SPA and IAI are cumulative.**
>
> ---
>
> > **Q4**: Sharing the codes?
>
> **A4**: We would like to make all codes publicly available, including detailed implementations, model training and experiment examples, along with user-friendly APIs. Given the substantial volume of implementation and experiment codes, we chose not to include them in the initial submission to allow more time for the improved organization and presentation of this project. **The code is now included in the supplementary materials**, and we will continue to refine the project to offer more comprehensive documentation and unified APIs.
>
> ---
>
> > **Q6**: How exactly is the mutual information computed for the continuous features and for regression labels?
>
> **A6**: In the implementation, we utilize the scikit-learn package to calculate mutual information. Specifically, for classification tasks, we employ the "feature_selection.mutual_info_classif" function, and for regression tasks, we use the "feature_selection.mutual_info_regression" function. Thank you for your reminder! We have depitcted the implementation in Appendix F.
>
> ---
>
> > **Q7**: Paper polishment.
>
> **A7**: Thank you for your careful suggestions. We will do careful proof-reading for the final version.
>
> Thanks again for your time and feedback. We've incorporated your suggestions into the manuscript, which is very helpful to improve the quality of our paper. We kindly request you to reconsider the ranking, considering the added ablation studies, additive studies and the practical benefits of ExcelFormer.
>
> ---
>
> Sincerely,
>
> Authors of paper 4888
>
> References:
>
> `[1]`  Revisiting deep learning models for tabular data. NeurIPS. 2021.

---

### Official Review · Reviewer_J7k8 · 2023-10-23

**Soundness:** 3 good
**Presentation:** 3 good
**Contribution:** 2 fair
**Rating:** 5
**Confidence:** 5

**Summary:**

The paper proposes a novel transformer architecture with two main suggestions:
- SEA: A semi-permeable attention block that masks the similarity scores from less informative features to the more informative ones, effectively blocking the transfer of information/representation from the less informative one.
- Interaction Attenuated Initialization: A rescaling of the variance of the weight initialization that in turn reduces the impact of SEA, which restricts feature interaction in the initial stages, making the proposed transformer architecture more non-rotationally invariant.

The authors then additionally propose two augmentation methods:
Hidden-Mix and Feat-Mix. One works by augmenting the data on the embedding space, while the other one works on the feature space by using the feature importance.

The authors combine the different components and one of the suggested augmentation methods at a time to yield an architecture that surpasses the baselines in 96 small-scale datasets and 21 large-scale datasets. The method outperforms the baselines without hyperparameter tuning and with hyperparameter tuning.

**Strengths:**

- The paper has a good structure.
- The authors propose quite a few interesting additions. The additions are ablated individually and the authors additionally show that the algorithm is more non-rotational invariant compared to the other transformer baseline.
- Experiments are extensive, a large number of datasets is considered and all the major baselines are included.

**Weaknesses:**

- The paper can be written better, typos exist here and there throughout the manuscript. (I will list a few of them in the questions section)
- The work should be self-contained and the "mutual information" should be described.
- In table 1, an interesting investigation would be how ExcelFormer would behave without any data augmentation (compared to the rest, not the ablation that is given) or how FTT would perform with the proposed augmentation approaches. I am additionally surprised that CatBoost performs worse compared to XGBoost consistently.
- No multi-class classification problems in the 96 datasets for the small-scale tabular datasets and only 4 datasets in the 21 large-scale datasets. 4 datasets in 117 datasets is an underrepresentation.
- Regarding the evaluation metrics, why would the authors use AUC for binary classification and ACC for multi-class classification? The latter would not be a good metric for imbalanced datasets.
- An ablation is given when mixup is used as data augmentation, however, I would also prefer to see cutmix usage as an ablation.
- Without code release, I find it difficult to trust the results, as unfortunately there exist a plethora of recent DL architectures that claim state-of-the-art performance (TabNet, Node, Saint, etc) [1][2][3] only to be debunked later on [4]. It is necessary to validate the proper setup of the baseline algorithms and to verify the results of the method.

[1] Arik, Sercan Ö., and Tomas Pfister. "Tabnet: Attentive interpretable tabular learning." Proceedings of the AAAI conference on artificial intelligence. Vol. 35. No. 8. 2021.

[2] Popov, Sergei, Stanislav Morozov, and Artem Babenko. "Neural Oblivious Decision Ensembles for Deep Learning on Tabular Data." International Conference on Learning Representations. 2019.

[3] Somepalli, Gowthami, et al. "SAINT: Improved Neural Networks for Tabular Data via Row Attention and Contrastive Pre-Training." NeurIPS 2022 First Table Representation Workshop. 2022.

[4] Shwartz-Ziv, Ravid, and Amitai Armon. "Tabular data: Deep learning is not all you need." Information Fusion 81 (2022): 84-90.

**Questions:**

- This presents a significant challenge and bottleneck in the broader adoption of neural networks on tasks involve tables. -> that involve tables*
- DNNs’ leanring procedure -> Learning *
- Section 2.1, the mask is defined as M, however, Equation 2 and the follow-up text continue with W
- Section 5.1, TabFPN -> TapPFN
- Section 5.2, indicating that applies hyperparameter finetuning onto EXCELFORMER can yield -> applying*

- Can you present the results where no augmentation is performed for ExcelFormer to analyze how it would perform against the other baselines? Can you provide the results where the proposed augmentation is applied to FTT?
- Can you include more multi-class classification problems in the used benchmarks and provide results?
- It would be interesting to see an ablation of the proposed augmentation methods against maybe cutmix or cutout to observe the overall improvement.

- **Is the EXCELFORMER more non-rotationally invariant and more noise insensitive?** In my perspective, an interesting addition would be to include the plain architecture of ExcelFormer in the investigation, then with every suggestion included one at a time (SEA, IAI), then both. This would show how the architecture gets more non-rotationally invariant as the different components are added compared to the beginning. Comparing against FTT is interesting, but it does not separate the impact of the overall differences in the architecture vs (SEA, IAI).
- I would urge the authors to provide the code to reproduce the results.

I am open to increasing my score if my concerns are addressed.

**Rebuttal Reply**
_____________________________________________________________________________________

I would like to thank the authors for their extensive reply. Below are my answers:

- **Regarding why previous SOTA has been debunked?**
I thank the authors for the explanation, although, I was of the same opinion initially.

**There are a few issues in the rebuttal from the authors, which I would like to point out:**

-  In Table 8 of the revised paper, with default hyperparameters, XGBoost in binary classification achieves a performance of 0 (failure case I would assume) which is surprising. Did the method fail since it does not have a competitive strategy for encoding categorical features like CatBoost? was one-hot encoding used?
- Typo on Table 8, the Excelformer with HID-MIX is highlighted instead of FEAT-MIX.

- The authors write that they use the XGBoost, CatBoost implementations/search spaces from (Gorishniy et al., 2021), however, the cited work compares transformers against GBDT methods, so the setup might be biased. I would suggest to use the implementations from papers that advocate that GBDT methods outperform DL methods.

- The goal of hyperparameter optimization (HPO) is to find better hyperparameter configurations compared to the default ones, while it seems that with HPO for the experiment related to multi-class classification in the majority of cases a method's performance drops, which again is surprising. This is not consistent with Table 3, where default hyperparameters have a worse performance.

- The aforementioned issue is additionally concerning in the case of Table 2, where the performance of CatBoost and XGBoost is given only with tuned hyperparameters. I would have liked to see the performance comparison where the aforementioned methods only use default hyperparameters. As an example in Table 8, CatBoost has a very strong performance with default hyperparameters.

- In Figure 5, it seems that FTT performs better with IAI and SPA. I could not easily find in what subset of datasets the ablation was done or if it was done on all datasets (in the latter case, the authors could further improve their preprocessing/backbone). This point does not take any novelty away from the components that the authors have proposed in the paper.

- It would be nice to run on already existing benchmarks, that are used in the community [1][2]. Although to be fair, the authors do consider an extensive amount of datasets, compared to previous works.

- I would strongly disagree with the use of accuracy for multi-class classification problems. Not only is it not consistent with binary classification problems, but it additionally does not capture the performance with imbalanced datasets.
- nRMSE is confusing in the provided results since higher values in magnitude are better. How are the authors calculating nRMSE? It seems more like a distance to the worst possible value.
- As a last note, the code should also include the baselines, to verify that they have been run properly.

[1] Salinas, D., & Erickson, N. (2023). TabRepo: A Large Scale Repository of Tabular Model Evaluations and its AutoML Applications. ArXiv, abs/2311.02971.

[2] Gijsbers, P., Bueno, M. L., Coors, S., LeDell, E., Poirier, S., Thomas, J., ... & Vanschoren, J. (2022). Amlb: an automl benchmark. arXiv preprint arXiv:2207.12560.


Lastly, I would again like to thank the authors for their extensive reply. I believe the work is stronger with the updated results, as such I am raising my score. Unfortunately, I would need a few more clarifications on the issues I raised to recommend acceptance.

---

> ### Author Response · Authors · 2023-11-21
> **To reviewer J7k8 (part 1)**
>
> Thank you for recognizing the good structure, comprehensive experiments, and the interesting aspects of our components. We trust that our responses will effectively address all your concerns.
>
> ----
>
> > **Q1**:
>
> > (i) Release codes?
>
> > (ii) Why previous works are debunked to be not a real sota?
>
> > (ii) Why propose several components?
>
> > (iii) Why XGboost outperforms Catboost?
>
> **A1**: Thank you for your thoughtful considerations. We would like to address the aforementioned questions together as they are closely related.
>
> (i) Certainly, we intend to make all codes publicly available, including detailed implementations, model training and experiment examples, along with some user-friendly APIs. We are confident that Excelformer can serve as a convenient and versatile solution for tabular data prediction. Given the substantial volume of implementation and experiment codes, we chose not to include them in the initial submission to allow more time for the improved organization and presentation of this project. **The code is now included in the supplementary materials**, and we will continue to refine the project to offer more comprehensive documentation and unified APIs.
>
> (ii) **Why previous SOTA be debunked?** Without intending any offense, we would like to explain why you find that recent approaches (e.g., TabNet, Node, SAINT) initially claimed state-of-the-art performance but were later "debunked". Actually, tabular datasets are highly diverse and heterogeneous. Unlike other fields (such as computer vision) where a single model can dominate most datasets, **different previous tabular data prediction models only excel on a part of datasets** (see Table 2: excluding Excelformer, the best models vary in different groups). In this paper, Excelformer tries to realize such dominance. Previous works ignored such dataset diversity, and most of them (e.g., TabNet, Node, SAINT) only validated their claims on limited datasets. Similarly, the conclusions in the paper titled *Tabular data: Deep learning is not all you need* may also not be entirely convincing, as it relied on only 11 datasets. Hence, in our paper, **we extensively utilized 130+ diverse datasets, and also presented results categorized by characteristics in Table 2**. See Table 2, one may observe that different previous methods exhibit preferences for distinct datasets, but Excelformer obtains top performances in all the groups.
>
> (iii) **Why propose several components?** Given the diversity of datasets in this field and the no-free-lunch principle, it is challenging to find a component widely applicable to all the dataset types (under default hyper-parameter setting). Following your suggestion, we compare the Excelformer (without data augmentation) against previous models, and found Excelformer (without data augmentation) achieves the best performance (see `Q4 & A4`). However, as an application-oriented research, we prioritize both model performance and user convenience. Thus, in this paper, we introduce those useful components, and we find they can make Excelformer:
>
> * exempt from hyper-parameter tuning (using default hyperparameters is very convenient in application!)
>
> * capable of robust performance across various types of datasets and tasks (see Table 2)
>
> We believe that the purpose of AI research is to facilitate practical applications. The combination of versatility and the convenience of not requiring hyperparameter tuning positions Excelformer as an excellent practical tool on tabular data prediction tasks, especially for users with limited computer science proficiency or computational resources.
>
> (iv) **Why XGBoost outperforms CatBoost?**
>
> * Firstly, in our experiments, **XGBoost does not consistently outperform CatBoost**. See Table 3, under default parameters, CatBoost outperforms XGBoost. Additionally, see Table 2, CatBoost performs better than XGBoost in classification and the case of #. Features $\ge$ 16. Besides, CatBoost typically outperforms XGboost in multi-class classification (see Table 7 of the revised version). **Such distinction is due to the diversity of datasets and tasks.**
>
> * A key contribution of CatBoost is reducing the workload for users in terms of feature engineering, visualization, and hyper-parameter tuning. This enables CatBoost to outperform XGBoost when using default parameters. However, after conducting extensive hyperparameter search for GBDTs (see Tables 6 and 7 - we provide more estimator candidates (4096) than previous works), we find that XGboost can outperform CatBoost on over 50% of datasets.
>
> **In summary, Excelformer is an application-oriented research. It achieves robust performance across diverse datasets using default hyperparameters. The proposed components work together and make Excelformer superior on the diversity of tabular data with almost no added cost.**

---

> > ### Author Response · Authors · 2023-11-22
> > **To reviewer J7k8 (part 2)**
> >
> > > **Q2**: The paper can be written better
> >
> > **A2**: Thank you for your careful suggestions. We have corrected those typos in the revised version and will do careful proofreading for the final version.
> >
> > ---
> >
> > > **Q3**: The work should be self-contained and the "mutual information" should be described.
> >
> > **A3**: In the implementation, we utilize the scikit-learn package to calculate mutual information. Specifically, for classification tasks, we employ the "feature_selection.mutual_info_classif" function, and for regression tasks, we use the "feature_selection.mutual_info_regression" function. Thank you for your reminder, and we have depitcted the implementation in the revised version.
> >
> > ---
> >
> > > **Q4**: Can you present the results where no augmentation is performed for ExcelFormer to analyze how it would perform against the other baselines? Can you provide the results where the proposed data augmentation is applied to FTT?
> >
> > **A4**: Thank you for your careful suggestion.
> >
> > **(1)** We compare ExcelFormer (without data augmentation) to previous works in Table 1 in the revised version, and **find Excelformer still performs best**. The detailed results are listed below:
> > | Model | XGboost (t) | Catboost (t) | FTT (t) | MLP (t) | DCN v2 (t) | AutoInt (t) | SAINT (t) | TransTab (d) | XTab (d) | Excelformer (t) |
> > |--|--|--|--|--|--|--|--|--|--|--|
> > | ave rank±std | 4.20±2.76 | 4.61±2.73 | 4.32±2.36 | 5.23±2.31 | 6.01±2.78 | 5.70±2.61 | 5.48±2.59 | 6.78±2.52 | 8.56±2.20 | **4.11±2.68**|
> >
> > However, it is worth noting that although Excelformer has surpassed previous works under fair hyperparameter fine-tuning, we would gently emphasize that, as an application-oriented research, we further introduce Feat-Mix and Hid-Mix so that:
> >
> > * The performance of Excelformer is further improved;
> >
> > * Excelformer can defeat other tuned methods with its default hyperparameters. This makes real-world practice very convenient and saves computational resources / time.
> >
> > **(2)** use the proposed data augmentations on FTT:
> >
> > Following your suggestion, we test how Feat-Mix and Hid-Mix performs with FTT, measured by performance rank. One can find that:
> >
> > **(i) Both Feat-Mix and Hid-Mix can improve the performances of FTT, which proves the benefits of our proposed data augmentation approaches.**
> >
> > **(ii) Feat-Mix and Hid-Mix on Excelformer respectively outperform those on FTT, which further proves the superiority of our Excelformer architecture.**
> >
> > The detailed results are presented below:
> >
> > | | FTT (no data augmentation) | FTT (Feat-Mix) | Excelformer (no data augmentation) | Excelformer (Feat-Mix) |
> > |-|:-:|:-:|:-:|:-:|
> > | ave rank ± std | 3.09 ± 1.16 | 2.59 ± 1.09 | 2.31 ± 1.00 | **2.01 ± 0.91** |
> >
> > | | FTT (no data augmentation) | FTT (Hid-Mix) | Excelformer (no data augmentation) | Excelformer (Hid-Mix) |
> > |-|:-:|:-:|:-:|:-:|
> > | ave rank ± std | 3.10 ± 1.13 | 2.34 ± 1.07 | 2.47 ± 1.07 | **2.09 ± 0.91** |
> >
> > Thank you again for your constructive suggestions and we will include these results in the final version.
> >
> > ---
> >
> > > **Q5**: No multi-class classification problems in the 96 datasets for the small-scale tabular datasets and only 4 datasets in the 21 large-scale datasets. 4 datasets in 117 datasets is an underrepresentation.
> >
> > Thank you for your constructive suggestion. We use fewer multi-class classification datasets because multi-class classification tasks themselves occur relatively infrequently among tabular data prediction tasks, especially on smaller-scale datasets. Some previous works have not included multi-class classification datasets`[1]`.
> >
> > Following your suggestion, we provide the results on 16 **additional multi-class classification datasets** (refer to Table 7 in the revised version), and we find that **Excelformer significantly performs best among various representative approaches**. We will included these results in the final version.
> >
> > | | XGb (t)       | Cat (t)  | FTT(t)  | MLP(t) | DCNv2(t)   | AutoInt(t)  | SAINT(t) | Excelformer(t)  |
> > |---|----|-|--|-|-|-|--|--|
> > | Rank($\pm$std) {$\downarrow$}    | 3.91$\pm$2.28     | 2.53$\pm$1.70     | 5.53$\pm$1.85    | 6.09$\pm$1.88    | 5.97$\pm$1.61    | 4.66$\pm$1.65    | 5.22$\pm$1.41    | **2.09$\pm$1.23** |
> >
> > ---
> >
> > > **Q6**: Why use ACC for multi-class classification?
> >
> > **A6**: ACC is the standard metric for multi-class classification tasks, which is widely used in previous tabular data prediction papers `[2,3,4]`. We try to follow these works so that offer easy reference for followers.
> >
> > ---
> >
> > References:
> >
> > `[1]` Why do tree-based models still outperform deep learning on typical tabular data? NeurIPS, 2022.
> >
> > `[2]` Revisiting deep learning models for tabular data. NeurIPS. 2021.
> >
> > `[3]` SAINT: Improved neural networks for tabular data via row attention and contrastive pre-training. 2021.
> >
> > `[4]` Tabnet: Attentive interpretable tabular learning. AAAI, 2021.

---

> ### Author Response · Authors · 2023-11-22
> **To reviewer J7k8 (part 3)**
>
> > **Q7**: It would be interesting to see an ablation of the proposed augmentation methods against maybe cutmix or cutout to observe the overall improvement.
>
> **A7**: Thank you for your suggestion. We compare the our data augmentation approaches with cutmix in two aspects:
>
> (1) We have added results of Excelformer architecture + cutmix into Figure 4, which shows that cutmix performs better than vanilla mixup but is still not as well as our proposed data augmentation approaches. To further analyze the difference between cutmix and Feat-Mix carefully, in Table 6 of revised version, we present the results of cutmix and Feat-Mix on several datasets. Theoretically, the difference between cutmix and feat-mix lies in the use of feature importance. Empirically, we find that:
>
> * In general, **Feat-Mix outperforms Cutmix** due to its ability to provide more accurate labels by considering feature importance.
>
> * When noise exists in the data, **Feat-Mix is less affected by the noise while cutmix is more heavily impacted**. This is clearly due to our use of feature importance when synthesizing new samples. Since tabular datasets have been shown to contain many uninformative features`[1]`, we believe Feat-Mix is a superior choice.
>
> The detailed results are as follow:
> | | Breast | Diabetes | Campus | cpu | fruitfly | yacht |
> |:-:|:-:|:-:|:-:|:-:|:-:|:-:|
> | CutMix (↑) | 0.702 | 0.822 | 0.972 | -102.06 | -16.19 | -3.59 |
> | Feat-Mix (↑) | **0.713** | **0.837** | **0.980** | **-79.10** | **-15.86** | **-0.83** |
> | CutMix (on noisy data) (↑) | 0.688 | 0.809 | 0.938 | -115.10 | -17.09 | -4.40 |
> | Feat-Mix (on noisy data) (↑) | **0.700** | **0.834** | **0.969** | **-74.56** | **-16.60** | **-0.89** |
> | $\Delta$ CutMix (↓) | 0.014 | 0.013 | 0.034 | 13.04 | 0.90 | 0.81 |
> | $\Delta$ Feat-Mix (↓) | **0.013** | **0.003** | **0.011** | **-4.54** | **0.74** | **0.06** |
>
> > **Q8**: An interesting addition would be to include the plain architecture of ExcelFormer in the investigation, then with every suggestion included one at a time (SEA, IAI), then both. This would show how the architecture gets more non-rotationally invariant as the different components are added compared to the beginning. Comparing against FTT is interesting, but it does not separate the impact of the overall differences in the architecture vs (SEA, IAI).
>
> **A8**: Thank you for your constructive suggestion. We conduct an additive study and an ablation study (see Fig. 5 of the revised version) to inspect the benefits introduced by SPA and IAI on breaking the rotational invariance property of DL models. We find that:
>
> * With SPA and IAI, our Excelformer shows better non-rotationally invariance, compared with previous works.
>
> * On the FTT backbone, the addition of SPA and IAI makes positive contributions to break the rotational invariance property.
>
> * On the Excelformer architecture, removing SPA and IAI weakens Excelformer's non-rotational invariance property.
>
> * **The effects of adding or removing SPA and IAI are cumulative.**
>
> Thank you once again for your valuable time and constructive feedback. We have taken your feedback into careful consideration and have made revisions to the manuscript accordingly. We kindly request you to reconsider the ranking, considering the improvements (especially the detailed ablation and additive studies)  made and the significant practical value that ExcelFormer offers.
>
> Sincerely,
>
> Authors of paper 4888
>
> ---
>
> References:
>
> `[1]` Why do tree-based models still outperform deep learning on typical tabular data? NeurIPS, 2022.

---

### Official Review · Reviewer_w8mp · 2023-10-31

**Soundness:** 3 good
**Presentation:** 3 good
**Contribution:** 3 good
**Rating:** 5
**Confidence:** 4

**Summary:**

This paper proposes a few modifications to the transformer
  architecture for small tabular data problems. The modifications are
  motivated by (1) the lack of rotational invariance in GBDTs and by (2) the
  efficacy of data augmentations in mainstream DL domains.

  To address (1), the authors propose:
  - semi-permeable attention (regular self-attention is masked such that more important features do not interact with less important features in self-attention)
  - interaction-attenuated initialization (initializing weights in semi-permeable attention with small values)

  To address (2) authors propose two variations of mixup tailored for tabular data problems:
  - Feat-Mix: swapping a random subset of features in two samples and mixing the labels taking feature's MI with the target into account
  - Hid-Mix: mixing channels after feature embedding and mixing labels proportionally, as in mixup

  With those changes, the proposed ExcelFormer outperforms deep-learning
  baselines (both traditional and more recent transformer-based models)
  and GBDTs in terms of average rank on 96 small (< 10000 samples) tabular problems.

**Strengths:**

- The paper is clearly and nicely written (both in overall structure and details in the technical details regarding proposed methods).
- It builds upon previous observations in its domain (tabular data) and proposes interesting "domain-specific" solutions to previously stated challenges/points for potential growth. For example, a large portion of the paper concerns with rotational invariance or the lack thereof as an inductive bias and.
- It obtains decent empirical results by integrating said solutions to the transformer architecture.

**Weaknesses:**

My concerns boil down to two things:

(1) **Using reduced rank as the main and only metric of model performance**. There are multiple problems I see with this approach to reporting the results, which keep me from agreeing with the ExcelFormer performance claims:
   - On this particular set of datasets DL models already perform on-par with GBDTs in terms of average rank (see FT-Transformer avg. rank), thus win over GBDT
   - The degree of improvement (in terms of the task metrics) is not quantifiable from the average rank. Did the ExcelFormer improved upon vanilla FT-Transformer by 10%, 50% in terms of AUC, neg. RMSE, ACC? The magnitude of the improvement is also important.

See also `[1]` regarding issues with comparing average ranks of multiple algorithms across multiple datasets.

I see that you provide all the results (albeit without standard deviations) for all models from Table 1, but this full table from the appendix is on the other side of the spectrum – too large to make generalizable conclusions. A more "zoomed in" view on performance would be very helpful. For example, you could provide metrics for DL baselines, GBDTs and ExcelFormer variants on datasets which were initially "won" by GBDT, but the changes introduced in ExcelFormer turned this around (I assume here that ExcelFormer is in essence a Transformer with potentially important domain-specific tweaks, comparison with MLP and FT-Transformer should be enough for a conclusion).

(2) **Limited ablations and comparisons to baselines**. The paper proposes a few architectural tweaks for a base transformer model: SPA instead of MHSA, IAI initialization in attention, new FFN block, new nonlinearity. With SPA and IAI highlighted as the more important ones. But the section with the ablation is rather short and lacking details regarding the setup, reporting only average rank performance. Could you provide a more detailed ablation and comparison to the vanilla transformer. For example:
- Transformer (no SPA, IAI, fancy embeddings and GLUs in FFN)
- Transformer + SPA
- Transformer + IAI
- Transformer + SPA + IAI
- ExcelFormer

A subset of datasets with metrics instead of ranks would be enough (see point 1).

For a second contribution - novel data augmentations, I believe they could be compared with baselines from pertaining on tabular data `[2,3,4]`, where resampling from marginal distributions for a set of columns was shown to be a decent augmentation. The results for the simplest possible setup (like MLP with all features linearly embedded – MLP-LR from `[5]`) with different augmentation strategies:
- Resample Augmentation
- Feat-Mix
- Hid-Mix
- Feat-Mix + Hid-Mix

would greatly improve the understanding of the efficacy of the proposed augmentations for tabular data.

In SPA and Feat-Mix, ExcelFormer uses mutual information. Could you discuss how different ways of estimating mutual information compare? It seems like a significant detail, but there are no mentions of this in the ablations or the experimental setup.

**References**:
- `[1]` Benavoli, Alessio, Giorgio Corani, and Francesca Mangili. "Should we really use post-hoc tests based on mean-ranks?." The Journal of Machine Learning Research 17.1 (2016): 152-161.
- `[2]` Bahri, Dara, et al. "Scarf: Self-supervised contrastive learning using random feature corruption." arXiv preprint arXiv:2106.15147 (2021).
- `[3]` Yoon, Jinsung, et al. "Vime: Extending the success of self-and semi-supervised learning to tabular domain." Advances in Neural Information Processing Systems 33 (2020): 11033-11043.
- `[4]` Rubachev, Ivan, et al. "Revisiting pretraining objectives for tabular deep learning." arXiv preprint arXiv:2207.03208 (2022).
- `[5]` Gorishniy, Yury, Ivan Rubachev, and Artem Babenko. "On embeddings for numerical features in tabular deep learning." Advances in Neural Information Processing Systems 35 (2022): 24991-25004.

**Questions:**

Technical details I'd like to clarify:
  - Could you provide details on how you compute mutual information, used in proposed augmentation and the attention module?
  - Could you provide more info on how ablations were run? You compare ablated variants to the fully tuned baseline, are the ablated variations also tuned?
  - How long were the models trained for? Was early stopping used during training? How the number of steps compare across deep models?
  - How ranks were calculated?

Other remarks:
- In the figure 3 hid-mix is called hidden-mix (only in the figure and nowhere else)
- The table with various datasets aggregations looks redundant in its current form. Not much interesting there besides TabPFN comparison. Not sure why grouping by classification vs regression and the number of continuous/categorical features should in differentiate general purpose methods. The results on the aggregated benchmark tell basically the same story: ExcelFormer is better than the baseline in terms of average rank. This space could be used to expand and address weaknesses (more ablations, more metrics).

Overall, I like the paper, and find the proposed architectural tweaks very interesting and important for the field.

I'm open to raise the score if my two concerns are addressed:

1. Results on multiple **challenging for DL datasets** where ExcelFormer significantly outperforms the DL competitors are demonstrated (not in ranks, but in raw metrics improved)
2. Comparisons for augmentations and ablations for SPA and IAI are presented (preferably on the datasets from point 1).

Looking forward to the discussion.

---

> ### Author Response · Authors · 2023-11-21
> **To Reviewer w8mp (part 1)**
>
> Thank you for acknowledging the clarity and significance of our work in this field and expressing your appreciation for our paper. We trust that our responses will effectively address all your concerns.
>
> ---
>
> > **Q1**: Using reduced rank as the main and only metric of model performance.
>
> **A1**: Thank you for your suggestions, and we acknowledge the limitations of mean-rank. We utilized the performance rank following previous works (e.g., `[1]`). Additionally, we compute **the average normalized scores** (introduced in `[2]`) to aggregate the model performances across datasets. The results are presented in Table 2, Table 7, Table 8 in the revised version. The average normalized scores linearly scale the model's performance, providing insights into the performance differences among models beyond just ranking. The definitions are presented in Appendix F.
> Interestingly, we observe that average normalized score and performance rank tend to offer consistent evaluations of the models. This consistency may be attributed to the substantial volume of datasets we employed. **Both performance rank and average normalized scores consistently demonstrate that our model outperforms previous works, on various datasets.**
>
> The detailed results are summarized below:
>
> **(1)** The performances on 96 small-scale datasets:
>
> * Binary-class Classification (51%; metric: Rank{$\downarrow$} (Average Normalized AUC {$\uparrow$}); best are in bold):
>
> | Model  | Excelformer | FTT (t)   | XGb (t)  | Cat (t)| MLP (t) | DCNv2 (t)     | AutoInt (t)    | SAINT (t)    | TransTab (d) | XTab (d)               | TabPFN (t)         |
> |-|-|--|-|-|-|--|--|--|--|--|--|
> | Setting: Hid-Mix (d) | **3.88 (0.79)**  | 4.88 (0.74)   | 5.97 (0.63)  | 5.77 (0.65)   | 6.61 (0.60)    | 6.38 (0.57)       | 6.63 (0.56)            | 6.07 (0.64)  | 6.31 (0.64)   | 9.50 (0.24)    | 4.01 (0.78) |
> | Setting: Mix Tuned   | **3.78 (0.80)**      | 4.91 (0.73)     | 5.95 (0.62)  | 5.79 (0.64)   | 6.60 (0.59)    | 6.39 (0.56)           | 6.71 (0.56)            | 6.10 (0.63)            | 6.37 (0.63)   | 9.46 (0.24)             | 3.95 (0.78)   |
>
> * Regression (49\%; metric: Rank{$\downarrow$} (Average Normalized nRMSE {$\uparrow$}); best are in bold):
>
> | Model | Excelformer  | FTT (t)  | XGb (t)  | Cat (t)   | MLP (t) | DCNv2 (t)  | AutoInt (t) | SAINT (t) | TransTab (d)  | XTab (d)  |
> |-|-|--|--|--|--|-|--|--|-|--|
> | Setting: Hid-Mix (d) | **3.81 (0.81)**  | 4.45 (0.78)  | 3.43 (0.83)      | 4.26 (0.80)  | 4.64 (0.74)   | 6.26 (0.52)  | 5.53 (0.65)  | 5.64 (0.66)  | 8.21 (0.33)   | 8.79 (0.20)   |
> | Setting: Mix Tuned   | **3.17 (0.86)**   | 4.49 (0.78)    | 3.53 (0.82)       | 4.28 (0.81)     | 4.74 (0.74)   | 6.32 (0.52)    | 5.68 (0.65)   | 5.72 (0.65)   | 8.23 (0.33)  | 8.83 (0.19)    |
>
> **(2)** On 21 larger-scale datasets, we compute the **normalized average scores of ACC, AUC, and nRMSE ($\uparrow$)** on multi-class classification datasets, binary-class classification datasets, and regression datasets, respectively. **It is evident that Excelformer performs best.**
>
> | Model (default hyperparameters)  | binclass    | regression  | multiclass|
> |-|-|-|-|
> | XGboost (d)  | 0 $\pm$ 0    | 0.470$\pm$0.400  | 0.218$\pm$ 0.400 |
> | Catboost (d) | 0.977$\pm$0.034  | 0.286$\pm$0.291  | 0.280$\pm$0.408  |
> | FTT (d)   | 0.807$\pm$0.397  | 0.613$\pm$0.313  | 0.763$\pm$0.163  |
> | ExcelFormer w/ Feat-Mix  (d) | 0.982$\pm$0.022  | 0.513$\pm$0.422  | 0.791$\pm$0.143  |
> | ExcelFormer w/ Hid-Mix (d) | **0.976**$\pm$0.030 | **0.825**$\pm$0.279 | **0.990**$\pm$0.020 |
>
> | Model (hyperparameter tuned) | binclass | regression| multiclass |
> |-|-|-|-|
> | XGboost (t) | 0.409$\pm$0.424  | 0.756$\pm$0.285  | 0.258$\pm$0.241  |
> | Catboost (t) | 0.281$\pm$0.365  | 0.276$\pm$0.385  | 0.095$\pm$0.135  |
> | FT-T (t)   | 0.316$\pm$0.387  | 0.428$\pm$0.398  | 0.596$\pm$0.425  |
> | ExcelFormer (Mix Tuned)| 0.526$\pm$0.413  | **0.865**$\pm$0.150 | 0.735$\pm$0.286  |
> | ExcelFormer (Fully Tuned) | **0.777**$\pm$0.393 | **0.865**$\pm$0.203 |**0.957**$\pm$0.085 |
>
> **(3)** Following the suggestions of the reviewer `J7k8`, we also provide the results on 16 **additional multi-class classification datasets** (refer to Table 7 in the revised version), and **both performance rank and average normalized ACC indicate that Excelformer performs best**:
>
> | | XGb (t)       | Cat (t)  | FTT(t)  | MLP(t) | DCNv2(t)   | AutoInt(t)  | SAINT(t) | Excelformer(t)  |
> |-|-|-|--|-|-|-|--|--|
> | Rank($\pm$std) {$\downarrow$}    | 3.91$\pm$2.28     | 2.53$\pm$1.70     | 5.53$\pm$1.85    | 6.09$\pm$1.88    | 5.97$\pm$1.61    | 4.66$\pm$1.65    | 5.22$\pm$1.41    | **2.09$\pm$1.23** |
> | average normalized ACC ($\pm$std) {$\uparrow$} | 0.65$\pm$0.35 | 0.82$\pm$0.27    | 0.39$\pm$0.33   | 0.25$\pm$0.31   | 0.33$\pm$0.35   | 0.50$\pm$0.31   | 0.42$\pm$0.29   | **0.85$\pm$0.24** |
>
> ---
>
> References:
>
> `[1]`XTab: Cross-table Pretraining for Tabular Transformers. ICML, 2023
>
> `[2]`Learning hyperparameter optimization initializations. DSAA, 2015

---

> ### Author Response · Authors · 2023-11-21
> **To Reviewer w8mp (part 2)**
>
> > **Q2**: Results on multiple challenging-for-DL datasets where ExcelFormer significantly outperforms the DL competitors should be demonstrated.
>
> **A2**: Thank you for your insightful suggestion. We agree that the tabular datasets are diverse, making parts of them specifically challenging for DL (i.e., GBDTs achieve relatively better performances while DL models obtain lower performances). In Table 2, we inspected the model performances on diverse types of tabular datasets / tasks. Additionally, in response to your suggestion, we introduced a subset known as "GBDT-best" datasets. The datasets in the "GBDT-best" group are selected based on: ignoring the results of Excelformer, XGBoost or CatBoost achieves the best performance. These results are included in the revised version (Table 2) with detailed analysis, and are also presented below.
>
> Though these datasets pose challenges for deep learning approaches, **we observed a reversal in performance with the changes introduced in ExcelFormer**:
>
> * Excelformer still outperforms GBDTs on "GBDT-best" classification datasets.
>
> * Excelformer surpasses all DL competitors and achieves competitive performances with CatBoost. Importantly, this doesn't imply that Excelformer lags behind XGboost in regression, since these datasets are GBDT-friendly. In fact, Excelformer outperforms GBDTs on regression tasks (please refer to `Q1 & A1`).
>
> * On those datasets initially "won" by GBDTs, Excelformer successfully outperforms GBDTs on 11 out of 15 GBDT-best classification datasets and outperforms GBDTs on 8 out of 22 GBDT-best regression datasets.
>
> Here we present the results on GBDT-best datasets (top 3, including GBDTs, are marked in bold):
>
> (1) Binclass Classification Tasks on GBDT-best datasets, measured by: Rank {$\downarrow$} (Ave Normalized AUC {$\uparrow$})
>
> | Model                  | Excelformer               | FTT (t)                | XGb (t)                | Cat (t)                | MLP (t)                | DCNv2 (t)              | AutoInt (t)            | SAINT (t)              | TransTab (d)           | XTab (d)               | TabPFN (t)             |
> |--|--|---|---|----|--|-|--|-|------------------------|------------------------|------------------------|
> | Setting: Hid-Mix (d)   | **4.63 (0.82)**  | 4.70 (0.79) | **3.50 (0.84)** | **3.73 (0.83)**  | 6.50 (0.65)            | 6.70 (0.64)            | 7.27 (0.53)            | 7.43 (0.63)            | 6.27 (0.69)            | 10.03 (0.16)           | 5.23 (0.71)            |
> | Setting: Mix Tuned    | **3.28 (0.88)**   | 4.63 (0.77) | **3.94 (0.82)**    | **3.81 (0.81)**  | 6.69 (0.64)            | 6.97 (0.63)            | 7.66 (0.53)            | 7.41 (0.63)            | 6.44 (0.68)            | 10.09 (0.15)           | 5.09 (0.72)            |
>
> (2) Regression Tasks on GBDT-best datasets, measured by: Rank {$\downarrow$} (Ave Normalized nRMSE {$\uparrow$})
>
> | Model                  | Excelformer               | FTT (t)                | XGb (t)                | Cat (t)                | MLP (t)                | DCNv2 (t)              | AutoInt (t)            | SAINT (t)              | TransTab (d)           | XTab (d)               |
> |------------------------|------------------------|------------------------|------------------------|------------------------|------------------------|------------------------|------------------------|------------------------|------------------------|------------------------|
> | Setting: Hid-Mix (d)   | **4.66 (0.70)**  | 4.95 **(0.70)** | **1.95 (0.89)**    | **3.11 (0.85)**    | 5.41 (0.64)| 6.50 (0.47)            | 5.86 (0.58)            | 6.00 (0.55)            | 8.14 (0.28)            | 8.41 (0.21)            |
> | Setting: Mix Tuned    | **3.18 (0.82)** | 5.09 (0.69) | **2.18 (0.87)** | **3.23 (0.85)**    | 5.55 (0.63)            | 6.64 (0.47)            | 6.18 (0.57)            | 6.18 (0.54)            | 8.27 (0.27)            | 8.50 (0.18)            |
>
> Thank you again for your constructive suggestion, which helps further validate the contributions of Excelformer. All the results with discussions are included in the revised version.

---

> ### Author Response · Authors · 2023-11-21
> **To Reviewer w8mp (part 3)**
>
> > **Q3**: The table with various datasets aggregations looks redundant...Not sure why grouping by classification vs regression and the number of continuous/categorical features ...
>
> **A3**: Thank you for your concern. We present performances on these subgroups separately because **tabular datasets exhibit significant diversity**, and thus different previous models only excel on specific portions of them (see Table 2: excluding Excelformer, the best models vary in different groups). This contrasts with domains like computer vision, where a single model can dominate most datasets. Excelformer aims to achieve a similar impact in tabular data prediction, but no previous works achieved this. **We have made some modifications and try to clarify this point in the introduction**.
>
> This diversity sparks ongoing debates within the community about the state-of-the-art (SOTA) model. Users often find model selection challenging, and reviewers `J7k8` also raises questions about the inconsistency in previous works being initially stated as SOTA and later "debunked". **Similar to drug studies analyzing various characteristics (such as race, age, pregnancy) to determine applicable scenarios, our Table 2 experiments aim to investigate the universality of compared models.** The proved universal applicability is a key advantage of Excelformer, not achieved by previous works, including GBDTs (as evident in Table 2, where GBDTs perform worse than FTT when the feature count is lower than 16). In essence, we argue that AI development aims to facilitate practical applications: the amalgamation of versatility and the convenience of not requiring hyperparameter tuning positions Excelformer as an excellent practical tool for tabular data prediction tasks, and Table 2 is helpful to users (especially those with limited computer science proficiency or computational resources) in model selection.
>
> Honestly, we are open to discussions and welcome further exploration of these considerations.
>
> ---
>
> > **Q4**: Limited ablations and comparisons to baselines. Comparisons for augmentations and ablations for SPA and IAI should be presented.
>
> **A4**: Thank you for your thoughtful question. We would like to clarify this points from two aspects: the selection of ablation study datasets; presenting more ablation results.
>
> * The selection of tabular datasets. **While some literature claims that GBDTs "dominate" tabular data prediction tasks, it's essential to clarify that this is not an absolute truth.** See Table 2, TabPFN often outperforms GBDTs in classification on small-scale datasets, and GBDTs may perform worse than other DL models when the feature count is lower than 16. Thus, **surpassing GBDTs is not the sole objective in building Excelformer**; beyond defeating GBDTs, we also aim to safeguard the advantageous niche of DL models (we have made some changes to clarify this point in the introduction section; thank you for bringing it to our attention). To simulate real-world applications, our ablation studies are based on all datasets (including those GBDT-best datasets).
>
> * More ablation study results.
>
> **(I)** Key Structure Components
>
> 1. Following your suggestion, we conduct additive study to examinate the performances of the key structure components on a vanilla Transformer (Tra). We find that all the components make positive contributions to tabular data prediction tasks, and the contributions are cumulative.
>
> | | Tra | Tra + IAI | Tra + SPA |Tra + SPA + IAI| Excelformer (no data augmentation) |
> |-|-|-|-|-|-|
> | binclass (Ave Norm AUC, ↑)| 0.232±0.36 | 0.683±0.37|0.289±0.37|0.753±0.33|0.775±0.30|
> | regression (Ave Norm nRMSE, ↑) |0.138±0.30| 0.738±0.25| 0.751±0.26| 0.766±0.35 |0.911±0.16|
>
> 2. Additionally, we perform an additive study and an ablation study (refer to Fig. 5) to examine the advantages introduced by SPA and IAI in disrupting the rotational invariance property of DL models. We find:
>
> * On the FTT backbone, the addition of SPA and IAI individually to some extent breaks the rotational invariance.
> * On the Excelformer architecture, removing SPA and IAI weakens Excelformer's rotational invariance property.
> * **The effects of adding or removing SPA and IAI are cumulative.**
>
> **(II)** Compare Hid-Mix, Feat-Mix, and Resample on MLP:
>
> The results are presented below (in Table 5 of the revised version). We observe that the performances of feature resampling fall between Feat-Mix and Hid-Mix. We notice that each approach excels on different datasets. We will provide more discussions on this aspect in the final version.
>
> || Resample | Feat-Mix | Hid-Mix | No data augmentation |
> |-|-|-|-|-|
> | Binclass (Ave Norm AUC, ↑) | 0.576±0.42 | 0.530±0.41 | 0.614±0.41 | 0.436±0.42 |
> | regression (Ave Norm nRMSE, ↑)|0.661±0.39|0.629±0.40 | 0.681±0.39 | 0.334±0.42 |
>
> All the additional results have been added into the revised version. Thank you for your constructive suggestion!

---

> ### Author Response · Authors · 2023-11-21
> **To Reviewer w8mp (part 4)**
>
> > **Q5**: How to compute mutual information?
>
> **A5**: In the implementation, we utilize the scikit-learn package to calculate mutual information. Specifically, for classification tasks, we employ the "feature_selection.mutual_info_classif" function, and for regression tasks, we use the "feature_selection.mutual_info_regression" function. Thank you for your reminder! We have depitcted the implementation in Appendix F.
>
> ---
>
> > **Q6**: You compare ablated variants to the fully tuned baseline, are the ablated variations also tuned?
>
> **A6**: Yes. For fair comparison, the ablated variations are tuned under the same setting.
>
> ---
>
> > **Q7**: How long were the models trained for? Was early stopping used during training? How the number of steps compare across deep models?
>
> **A7**: We implemented early stopping with a patience of 32, and the maximum epoch for Excelformer was set to 500. To ensure a comprehensive tuning of XGBoost and CatBoost for optimal performance, we conducted 500 tuning iterations and increased the number of estimators to 4096 (most previous works used less than 2000). Excelformer underwent fine-tuning for 50 iterations across all datasets. For other deep learning approaches, on large-scale datasets, we followed the settings outlined in `[1]` or `[2]`. On small-scale datasets, all deep learning approaches were tuned for 50 iterations, as we found this to be sufficient for small-scale datasets and is fair among DL approaches. We did not set a time limit because fine-tuning on small-scale datasets does not consume an excessive amount of time. We have included these information in the revised version, thank you!
>
> > **Q8**: How ranks are computed?
>
> **A8**: We performed 5 runs with different random seeds and calculated the average results for each dataset. Additionally, we computed the overall rank across datasets for comparison. Average rank is given to tied values. These details were included in Appendix F of the revised version. Thank you!
>
> > **Q9**: In the figure 3 hid-mix is called hidden-mix (only in the figure and nowhere else)
>
> **A9**: Thank you for bringing this to our attention. We have revised Figure 3 in the updated version.
>
> **Thank you again for your time and your constructive feedback, which we've carefully considered and incorporated into the revised manuscript.** We kindly request a reconsideration of the ranking, taking into account the improvements made and the substantial practical value of ExcelFormer.
>
>
> Sincerely,
>
> Authors of paper 4888
>
> ---
>
> References:
>
> `[1]`Revisiting deep learning models for tabular data. NeurIPS, 2021.
>
> `[2]`Why do tree-based models still outperform deep learning on typical tabular data? NeurIPS, 2022.

---

### Meta-Review · Area_Chair_ZUmK · 2023-12-02

**Metareview:**

The paper addresses neural classification architectures for tabular datasets. The authors propose a novel attention mechanism and two data augmentation approaches. Overall, the paper demonstrates interesting improvements in terms of predictive performance.

However, the reviewers highlighted limitations in terms of ablating the various ingredients of the method, because the paper introduces multiple contributions at once. The authors did provide ample empirical evidence with an elaborate rebuttal. Based on the new rebuttal results, the proposed technique seems indeed to exhibit a strong empirical performance. Unfortunately, I believe the vast amount of new experimental results will need to be integrated into the main manuscript, and placed in a coherent story. The experimental protocol of the new rebuttal experiments needs to be clarified in detail. Such major changes require a new thorough review process. As a result, I vote for rejecting the paper in its current form. However, I do acknowledge the merits of the paper and encourage the authors to re-organize the experimental protocol, integrate the new results, and resubmit to the next conference.

**Justification For Why Not Higher Score:**

The new rebuttal results need to be integrated into the main manuscript and thoroughly reviewed.

**Justification For Why Not Lower Score:**

N/A

---

### Decision · Program_Chairs · 2024-01-16

Reject